| **Open Peer Review** | Genetics and Molecular Biology | Methods and Protocols

# High-throughput chemical genomic screening: a step-by-step workflow from plate to phenotype

Georgia Williams,[1,2] Huda Ahmad,[1] Susan Sutherland,[1] James Haycocks,[3] Sam Benedict,[3] Adam J. Hart,[3] Hannah H. Doherty,[1] Rudi Sullivan,[1] Micheal Alao,[1] Xuyu Ma,[3] Qianhui Xu,[3] Jack Bryant,[4] Monika Glinkowska,[5] Peter Banks,[3] Patrick Moynihan,[1] Mathew T. Milner,[2] Danesh Moradigaravand,[2] Manuel Banzhaf[1,3]

**ABSTRACT** High-throughput chemical genomics uses phenotypic profiling of strain libraries under defined chemical and environmental conditions to identify gene functions. This approach enables the mapping of biological pathways and can potentially highlight drug targets. Chemical genomic data sets have been springboards for numerous hypothesis-driven research projects, with direct implications for antimicrobial resistance and clinical outcomes. High-throughput phenotypic profiles are valuable tools for enriching microbial sequence data with functional annotations and benefiting the broader scientific community. This work provides a step-by-step guide for conducting chemical genomics screens from start to finish.

**IMPORTANCE** Chemical genomic screening is a powerful systems biology approach for linking gene function to phenotype under diverse chemical and environmental stressors. However, its broader use in microbial research has been limited by the lack of standardized, reproducible workflows. Our study introduces a scalable, end-to-end protocol that integrates experimental, imaging, and computational steps into a cohesive framework for high-throughput screening across a range of microbial species. This enables researchers to generate consistent, high-quality phenotypic data suitable for large-scale analyses. The protocol supports systematic exploration of gene-environment interactions, microbial stress responses, and antimicrobial resistance. Its adaptability and troubleshooting guidance make it especially useful for groups working in microbiome research, synthetic biology, and microbial community studies. By bridging benchwork and computational analysis, this workflow expands the technical toolkit available to microbial systems biologists. Our work contributes to the development of robust methods for functional genomics and supports the core mission of mSystems to advance microbial systems biology.

**KEYWORDS** chemical genomics, bioinformatics, genomics, systems microbiology

**Peer Reviewer** Xue Liu, Shenzhen University Health Science Center, Shenzhen, China

Address correspondence to Manuel Banzhaf, manuel.banzhaf@newcastle.ac.uk, Danesh Moradigaravand, danesh.moradigaravand@kaust.edu.sa, Mathew T. Milner, mathew.milner@kaust.edu.sa, or Georgia Williams, GXW022@student.bham.ac.uk.

Georgia Williams, Huda Ahmad, and Susan Sutherland contributed equally to this article. The author order was determined by their contribution to the article.

The authors declare no conflict of interest.

See the funding table on p. 24.

The rapid expansion of sequenced microbial genomes over recent decades has transformed our understanding of microbial diversity but also highlighted a major gap: the challenge of assigning functional annotation to genes.

Chemical genomic screens provide a systematic approach to evaluate the phenotypic impact of chemical or environmental perturbations on single-gene mutant libraries. Using diverse phenotypes as measurable outputs, chemical genomics screens systematically map colony size as a proxy for fitness, color uptake to quantify biofilm formation, and alterations in colony topology to assess biofilm morphology (1).

Chemical genomic screens can be broadly categorized into arrayed library-based and sequencing-based pooled screening approaches (2, 3). In the arrayed screen, which this protocol describes, individual mutant strains are arrayed on solid or in

liquid media to enable direct phenotypic measurements (e.g., colony size, morphology, biofilm formation, etc.) at the strain level. In contrast, pooled screening approaches, such as Tn-seq and CRISPRi-seq, involve mixed mutant populations being exposed to selective pressures, followed by next-generation sequencing to track changes in mutant prevalence (4, 5). Pooled screens offer higher throughput and are often more cost-effective per condition, particularly for genome-wide analysis across single conditions (6). However, arrayed screens allow for precise assessment of strain-specific responses, including morphological traits, biofilm formation (1), and minimum inhibitory concentration (MIC) (7), and are in general better suited to study essential genes (e.g., CRISPRi libraries) (8). In addition, arrayed screens allow for visual inspection, real-time imaging, and easier follow-up experiments on individual strains.

While individual observations can be analyzed independently to establish a functional link between a specific condition and a corresponding genetic perturbation, the true utility of chemical genomic screens lies in their ability to generate phenotypic profiles. Hierarchical clustering of all phenotypic profiles constructs a genome-wide epistasis map, identifying similarity patterns and facilitating the functional grouping of genes and stress conditions (9). Chemical genomic screens have led to many important biological discoveries. While some foundational high-throughput screening studies, such as those in *Saccharomyces cerevisiae* (10, 11) used genetic interaction profiling rather than chemical genomics, they helped to establish core methodologies that informed later chemical genomic approaches. Notable chemical genomics studies include work in *S. cerevisiae* (12), *Acinetobacter baylyi* (13), *Pseudomonas aeruginosa* (14), *Escherichia coli* (15–19), *Bacillus subtilis* (20), and *Salmonella enterica* (21). However, there is currently no dedicated protocol that assists research groups in establishing their chemical genomic studies. This guide outlines a step-by-step workflow to help scientists establish chemical genomics screens.

## Foreword

The protocol outlined in this paper serves as a comprehensive, step-by-step guide to conduct chemical genomic screens in bacteria on solid medium. The methodology presented is designed to be broadly applicable to all bacterial species with limited changes to the protocol ensuring a high reproducibility. We have highlighted some species-specific observations based on our previous experiences (see "Species-Specific Considerations") and addressed common troubleshooting challenges (see "Troubleshooting"). As each experimental step in this workflow encompasses its own materials, methods, outcomes, and interpretation, we have chosen to present the protocol in an integrated format that combines the Materials, Methods, Results, and Discussion into a single continuous section.

## Content

1. **Plate pouring**. Consistency in plate pouring is essential for chemical genomic screens as it ensures uniform colony growth, accurate phenotypic observations, and reproducibility by providing consistent surface conditions and even distribution of stress conditions. Proper techniques also minimize contamination and variability, improving the reliability of screening results.
2. **Source plate production**. Source plates serve as templates, ensuring consistent and reproducible sample distribution. They enable high-throughput screening by allowing transfer of strains to condition plates, reducing contamination and variability while preserving genetic diversity.
3. **Pre-testing**. Pre-testing helps validate assay conditions, optimize protocols, and identify potential issues before large-scale screening. This reduces errors, ensures

reproducibility, and improves the reliability of results by confirming that the screen functions as intended.

4. **Screening methodology**. This section describes the methodology used for stamping source plates to condition plates. It is designed specifically for the image analysis software, Iris (1). Additionally, the protocol includes guidance on biofilm-specific screens (see "Biofilm screening"), a research area of growing importance given the role of biofilms in bacterial persistence and resistance.

5. **Computational analysis**. Iris (see "Taking images and Iris") and phenotypic analysis software, ChemGAPP (see "Phenotypic analysis and visualisation") (9) to ensure that phenotypic data are accurately quantified and interpreted.

## Equipment specificity

This protocol has been developed for use with the S&P Robotics BM series pinning robot available in our laboratory. While the core workflow and screening principles are broadly applicable, specific steps, such as pinning technique, sterilization procedures, and plate handling, may differ when using alternative platforms (e.g., Singer ROTOR HDA, manual pinning tools, etc.). Users are advised to adapt these steps based on their own equipment and consult relevant manufacturer protocols. Equipment-dependent considerations and adaptations are highlighted where appropriate throughout this workflow.

## MATERIALS AND METHODS

### Pre-screening

#### *Plate pouring*

*Purpose*

Plate quality affects the accuracy and reproducibility of the generated data. Plates need to be equally poured (without contamination) and dried to avoid introducing systematic pinning biases caused by agar bubbles, over-dried plates, wet plates, etc. Therefore, we recommend taking great care when pouring agar plates.

*Materials*

- VWR Single Well Plates (Cat no: 734-2977) (culture area, 97 $cm^2$, recommended working volume 35 mL, well volume, 73 mL) or any other single-well plate.
- Suitably sized stripette for drawing up media to correct volumes.
  - Alternatively, a suitably sized vessel can be used instead for directly pouring into plates.
- Glass bottles (autoclavable).
- Base growth medium appropriate for your bacterial species.
- Agar—to achieve 2% (w/v) plates.
- Distilled (type 2) water.
- The experimenter's choice of stress conditions, e.g., antibiotics (see 'Pre-testing").

*Method*

1. **Prepare growth media**: Use 2% agar and ensure all components are fully dissolved and mixed with a magnetic stirrer. Adjust pH as needed (Table 1). Autoclave the media before use.

**TABLE 1** Plate pouring challenges and solutions

| Challenge | Issue | Solution |
|---|---|---|
| Plate labeling | Plate naming inconsistency can impact the ease of tracking screening progress and labels on the bottom of plates can impact the image analysis. | Label all plates with a consistent system throughout the screen on the plate's side |
| Reducing variation | Plate batch can have an impact on the observed colonies. | Plate batches should be recorded and accounted for in the results. |
| Consideration of growth medium agar temperature | Keeping the growth medium agar at room temperature for an extended period will make it begin to solidify unevenly, which results in clumps of agar forming in newly poured plates. | Place autoclaved agar immediately into a 55°C–65°C water bath or incubator. (Note that some condition additives may be temperature sensitive and may need to be added at a different temperature.) |
| Plate drying | Good plate drying can be time-consuming at room temperature. | The drying process can be sped up by drying plates under a steady flow of air within a laminar flow hood. Be mindful, as drying the plates out too much often introduces pinning artifacts. |
| Plate storage | Plates may dry out if not stored correctly when longer-term storage is required. | Plates can be stored safely in a fridge (4°C–8°C) for up to 4 weeks. However, this depends on the condition (e.g., some additives may precipitate at 4°C). |
| Setting plates | Biased plate surfaces will lead to inconsistencies in colony transfer during pinning. | Ensure the surface that you pour plates on is level. |
| Plate drying during incubation | Plates can dry out during incubation based on agar thickness and the organism on the plates' growth time. | For faster-growing organisms, agar volumes of 40 mL are appropriate (if using VWR one-well plates). For slower-growing organisms, such as *Mycobacterium bovis* BCG, thicker plates are recommended (45–50 mL agar volumes) to prevent plates from completely drying during incubation. Incubators with humidity are also recommended for slower-growing organisms to reduce the rate of evaporation. |

2. **Add chemical conditions**: Add chemical additives (e.g., antibiotics, dyes, etc.) once the agar reaches an appropriate temperature. Mix thoroughly to ensure even distribution.
3. **Pour plates aseptically**: Pour agar into pre-labelled plates using sterile technique (e.g., Bunsen burner or safety cabinet). Use a consistent volume suited to the plate type, e.g., 40 mL for VWR one-well plates (~2/3 full) to reduce dehydration.
4. **Dry plates before use**: Allow plates to fully solidify for a minimum of 3–4 h at room temperature before inverting. After this initial drying period, invert the plates and dry them at room temperature for an additional 16 h to remove excess moisture.

## Source plates

### Purpose

Plates are split into:

1. **Library plates:** These plates are the original stocks of bacteria and are typically stored as liquid culture. Usually, for long-term storage of the bacteria, they are mixed with glycerol or DMSO to act as cryoprotectant and stored at −80°C.
2. **Source plates:** These plates contain growth medium that is either in solid or liquid format depending on the strains of interest. They are generated from liquid culture (library plates) and are the principal plates to be pinned/stamped out onto condition plates (solid or liquid plates).

In chemical genomics screens, source plates are replicated onto condition plates to study microbial strains (Fig. 1A). As all transfers originate from source plates, their quality

is crucial. Strains must be well-grown and accurately transferred, as issues will affect all condition plates (Tables 2 and 3).

Avoid under- or overgrown colonies when culturing bacteria, as both can affect replication quality. Undergrowth may limit transfers, while overgrowth can cause uneven pinning and downstream artifacts. An example of an acceptable source plate is shown in Fig. 1B (For troubleshooting: see "Troubleshooting" and Table 7). To select optimal source plates, we recommend pinning extra copies and then choosing the best.

### Materials

- Prepared growth media plates from "Plate pouring," containing no stressor.
- Library plates.
- Pinning robot.
- 70% ethanol for cleaning the robot ("Screening").

### Method

1. **Thaw and prepare library plates**: Fully thaw library plates before transfer. Centrifuge at low speed (~250×*g* for 1 min) to collect contents before removing the seal.
2. **Create source plates**: Use a pinning robot or hand-pinning tool to transfer glycerol-stored strains from library plates onto growth media, generating new source plates. Multiple library plates can be combined into various screening

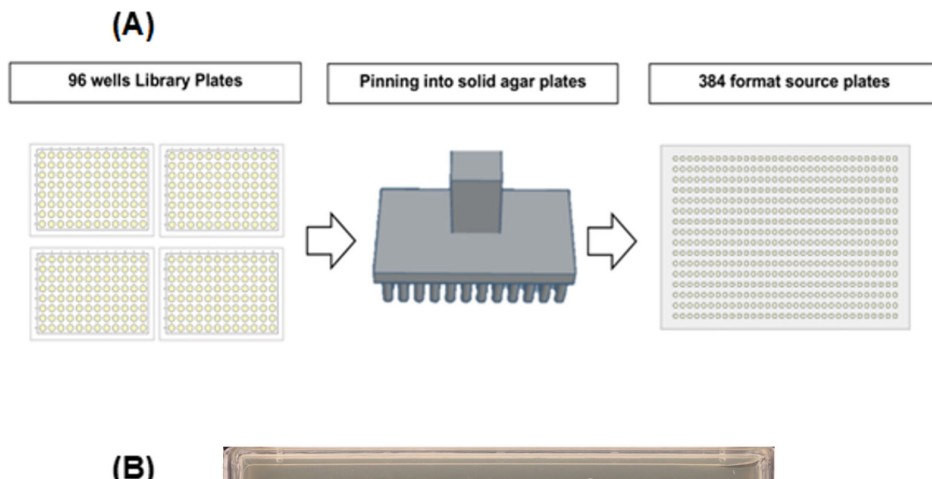

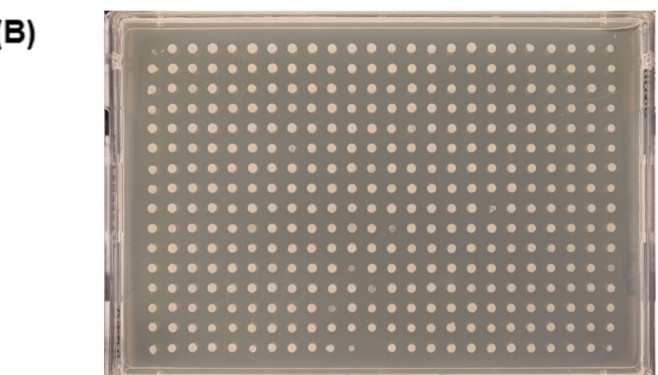

**FIG 1** Source plate preparation. (A) Library plates with strains can be arranged in a varied format, the figure shown shows four 96 well plates that are pinned into source plates and arrayed in a 384-source plate format. (B) Image taken to show a representative example of a source plate. Good-quality source plates can be reliably replicated to produce condition plates.

**TABLE 2** Source plates challenges and solutions

| Challenge | Issue | Solution |
|---|---|---|
| Defrosting | Insufficient defrosting of plates and centrifugation of plates prior to plate pinning. This process ensures that a sufficient amount of bacteria is pinned onto the source plates and reduces the risk of cross-contamination upon removal of plate seals. | Ensure that any original frozen libraries in library plates stored at −80°C are thoroughly defrosted at room temperature and then centrifuged prior to being pinned. |
| Viability of source plates | Ensure that species are still viable before commencing the screen. Once prepared, they can remain viable for up to four weeks when stored at 4°C–7°C. | Generally, source plates can remain viable for up to four weeks when stored at 4°C–7°C. After this period, re-pinning is required; however, this is species-dependent. |
| Replicate plate copies | Ensure to create sufficient replicate plate copies to avoid having to undertake unnecessary repeats. | For all stored library plates, it is strongly advised to create at least two identical copies of these plates and store these appropriately. This step can be undertaken manually or using an automated liquid handler. |
| Minimizing physiological variability | Physiological variability during cell transfer can lead to artifacts in the results. | See "Source plates - Minimizing Physiological Variability." |

formats (Fig. 2). Additional information related to creating and storing source plates can be found in Table 2.

3. **Maintain sterility and optimize growth**: Ensure sterile technique during transfers and use sealing films suited to the plate type. During media transfer, aspirate any residual media from pipette tips. Adjust growth time and temperature based on the organism and plate format, for example, 1,536-format plates typically require shorter incubation than 96-format plates due to tighter colony spacing.

## Minimizing physiological variability

To ensure reproducibility and to minimize any artifacts during the transfer of strains from source plates to condition plates, it is essential to standardize bacterial growth conditions and include potential controls for potential variability. Below are recommended strategies to reduce physiological variation.

1. Synchronization of growth phase.

When generating source plates from liquid library plates, ensure all cultures are in the same physiological growth phase by:

a. Thawing and incubating library plates uniformly.
b. Using consistent incubation times and conditions (e.g. temperature, shaking speed, etc).
c. Monitoring OD600 of pre-cultures where applicable to standardize the starting cell density.

2. Consistent pre-culture conditions.

Pre-culture media, temperature, and shaking conditions should be identical across all library plates used for a screen. Variability in these parameters can affect bacterial physiology and impact transfer efficiency.

3. Use of internal plate controls.

Ensure internal controls are incorporated within each source and condition plate by:

a.  Spotting known reference strains (e.g., wild type) at fixed and interleaved positions across plates. This will allow for internal normalization and tracking of plate-to-plate variability.

4.  Spotting consistency metrics.

Assess the consistency of the spotting process by:

a.  Imaging source plates post-pinning to verify uniform colony size and distribution.
b.  Monitoring the coefficient of variation (CV) for colony sizes across technical replicates.
c.  Establishing thresholds (e.g., <10% CV) to accept/reject plates prior to screening.

Implementing these practices enhances the robustness of chemical genomic screens by ensuring that variability is due to the experimental conditions and not physiological differences between cultures.

### *Pre-testing*

#### *Purpose*

Pre-testing ensures that chemical genomic screens are cost and time efficient. It reduces the need for repeat experiments by optimizing the condition concentrations and incubation times to obtain the dynamic growth range.

The dynamic growth range allows for the accurate assessment of how a condition influences phenotype, describing an endpoint that enables clear differentiation in colony size between sensitive and resistant strains, helping determine MICs and partial inhibition. The optimal range for each condition can vary depending on the incubation time and the conditions' effect on growth, as both factors together determine the end point colony size (Fig. 3; Tables 1 and 2).

*Note:* While this protocol emphasizes chemical perturbations (e.g., antibiotics, dyes, etc.) it can also be adapted for environmental conditions, such as heat stress, pH shifts, or osmotic stress. Unlike chemical agents, environmental conditions can also be applied externally, e.g., by adjusting incubation temperature or atmospheric conditions, as opposed to being incorporated directly into the agar. Users should validate these environmental conditions during pre-testing to ensure consistent application across all plates and minimize artifacts.

After pinning, plates are incubated based on the organism and stress conditions (see "Species-specific considerations," Table 8). As chemical genomic screens use a single imaging endpoint, users must determine the optimal time by evaluating key parameters to ensure data can be captured (e.g., via Iris). These include:

a.  Allowing sufficient time for colonies to grow, ensuring effective data collection without over- or undergrowth (Fig. 3C).
b.  Ensuring adequate spacing between colonies to avoid overlap.
c.  Avoiding edge effects and cross-contamination (Fig. 4).

#### *Materials*

- All produced condition plates (see "Plate pouring").
- Source plates (see "Source plates").
- Incubator.

**TABLE 3** Pre-testing challenges and solutions

| Challenge | Issue | Solution |
|---|---|---|
| Determining stress condition concentrations | The optimum concentration needs to be investigated to determine a suitable concentration for the entire microbial library present on the condition plates. | Consult literature data for conditions, examples include (22) for *Escherichia coli* along with *S.* Typhimurium and *P. aeruginosa*, while (17) focused specifically on *Escherichia coli*. Websites, such as EUCAST (23), provide antimicrobial susceptibility data for a broad range of organisms and can help identify appropriate antimicrobial concentrations. |
| Serial dilutions | The MIC must be determined either by literature consultation or through serial dilutions. | For further MIC determination, prepare a serial dilution of the chemical using a 10-fold or 2-fold dilution series on 3–4 conditions to stimulate the dynamic range. The final concentrations must span across a range expected to include the MIC for the species you're testing. Use pre-testing plates for this purpose. |
| Determining incubation time | Ensure that incubation time is optimized for specific species being screened. | If this is the first time performing the procedure on a screening plate, check the plates hourly to determine the optimal incubation time for the strains. This will establish a baseline incubation period, which can then be applied consistently for all strains in the screen. |
| Stamping source plates to condition plates | Over-stamping a source plate can dilute the transferred cell mass, leading to inaccurate results. | Determining the number of condition plates per source plate is important as the source plate serves as a biological replicate. An initial test screen using one source plate and multiple condition plates is recommended. Typically, 10–15 plates can be stamped from a single source plate (see Table 8). |
| Plate format for biofilms | Growth media plates may not be suitable for strains forming biofilms (see "Biofilm screening"). | Typically, solid source plate to solid screening plate stamping is conducted to ensure a consistent amount of cell mass is transferred for each strain. In cases where growth media plates are not suitable for specific species, liquid culture can be used instead as a source plate: thaw glycerol stocks, inoculate a fresh liquid culture plate to saturation, and then stamp directly onto condition plates to maintain uniform cell transfer. |

## Method

1. **Optimize pre-testing**: Determine stress condition concentrations that allow sufficient growth with visible phenotypic differences (see Tables 1 and 3). If unclear, use MIC broth microdilution for guidance (see Table 3).
2. **Standardize format**: Use consistent plate formats and growth media across the screen to ensure comparability.
3. **Refine incubation**: Adjust incubation time based on organism and condition stress level. Use control plates (no stress) to benchmark growth rates and compare with condition plates to define a dynamic range. For example, if control growth occurs at 9 h, low and high stress may delay growth to 10 or 12 h respectively. Ensure colonies remain resolvable for Iris analysis.
4. **Prevent artifacts**: Control for issues like colony overlap, edge effects, and contamination to improve data quality (Fig. 4).

## Screening

### Purpose

This section outlines a general screening design for chemical genomic screens and describes the process of pinning from source plates onto condition plates (Fig. 5). We recommend using a minimum of 3–4 replicates per condition, with each replicate coming from an independent source plate (biological replicate). Several tools are available to perform screens; this is discussed further in "Screening design."

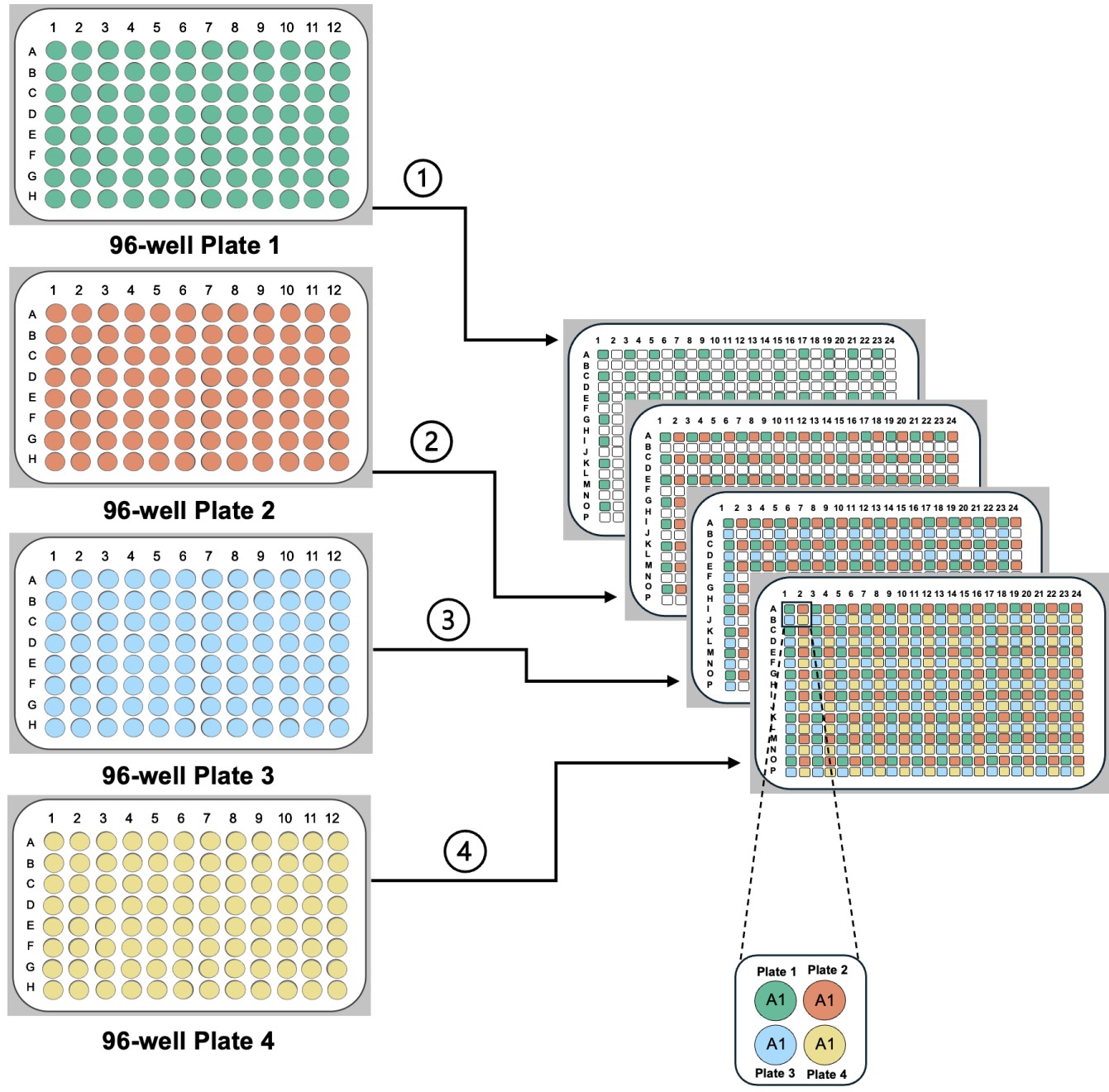

**FIG 2** Illustration of a 384 constructed source plate from four 96-well library plates. Colors represent strains from each of the 96-well plates and subsequent layout on a 384-well plate.

## Materials

- A pinning head (a tool used to replicate plates typically in 96, 384, or 1,536 configurations) (see "Screening design").
- 70% ethanol for sterilizing.
- Condition plates containing conditions (see "Pre-testing").
- Source plates (see "Source plates").
- Static incubator.

### Method

1. **Prepare condition and source plates**: Inspect all plates for defects (e.g., bubbles, uneven surfaces) and if they have been contaminated. Discard all plates that fail this final quality control. To avoid pinning artifacts, we recommend bringing source plates to room temperature (Table 2).
2. **Ensure sterility**: In a sterile environment, ensure that pinning components are clean (Table 4).
3. **Pinning procedure**: Select an appropriate pinning protocol (Table 8 for replication limits per source plate). A single source plate can be used to stamp multiple condition plates (Table 2). Pinning involves the following:
   a. Aligning the pinning head over each strain on the source plate.
   b. Gently lowering it to pick up a small, consistent amount of material.
   c. Transferring material by lightly touching the pins to the condition plate without piercing the agar.
4. **Sterilize between source plates**: When switching source plates, sterilize the pinning head with 70% ethanol and allow to air dry before repeating the pinning process (Table 4). *Note:* The reference to non-dispensable pinning pads applies specifically to the robot model used in this protocol. Many laboratories may use dispensable (single-use) pinning pads in their chemical genomics workflows, which might require different sterilization or handling procedures.
5. **Incubation**: Place condition plates in a static incubator at the appropriate temperature and duration for the organism used (see "Pre-testing").
6. **Imaging**: Once colonies reach an appropriate size for image analysis (e.g., via Iris), remove plates from the incubator and proceed with imaging.

## Imaging and data analysis

### Taking images and Iris

#### Purpose

This section outlines best practices for imaging plates and analyzing phenotypic changes such as colony size and opacity with Iris, ensuring plates are suitable for downstream phenotypic analysis (1). Figure 6 highlights this process.

#### Materials

- Camera: A built-in fixed 18-megapixel Canon EOS Rebel T3i camera (Canon) is used within our group; however, generally, any suitable camera to take plate images can be used. Refer to the robot manufacturers' protocols for guidance on specific types of cameras that are suitable for the robot type.
- Ensure to maintain a fixed distance between the camera and plates as well as consistent lighting, preferably with a lightbox. This minimizes variability across images.
- Software: S&*P* Imager (S&*P* Robotics, version 2.0.1.0) and EOS Utility (Canon, version 2.10.0.0) is used within our group.

Download and install Iris (v0.9.7, ) for image analysis. Iris is a scalable and flexible platform capable of quantifying multiple features of high-density arrayed microbial colonies. This open-source software supports a wide range of microbial species and includes add-on features for compatibility with both low-throughput and kinetic data (1). Alternative image analysis software programs are also available, such as ScreenMill (24) and gitter (25); however, these programs would need to be validated to ensure that they can be used with specific screen setups.

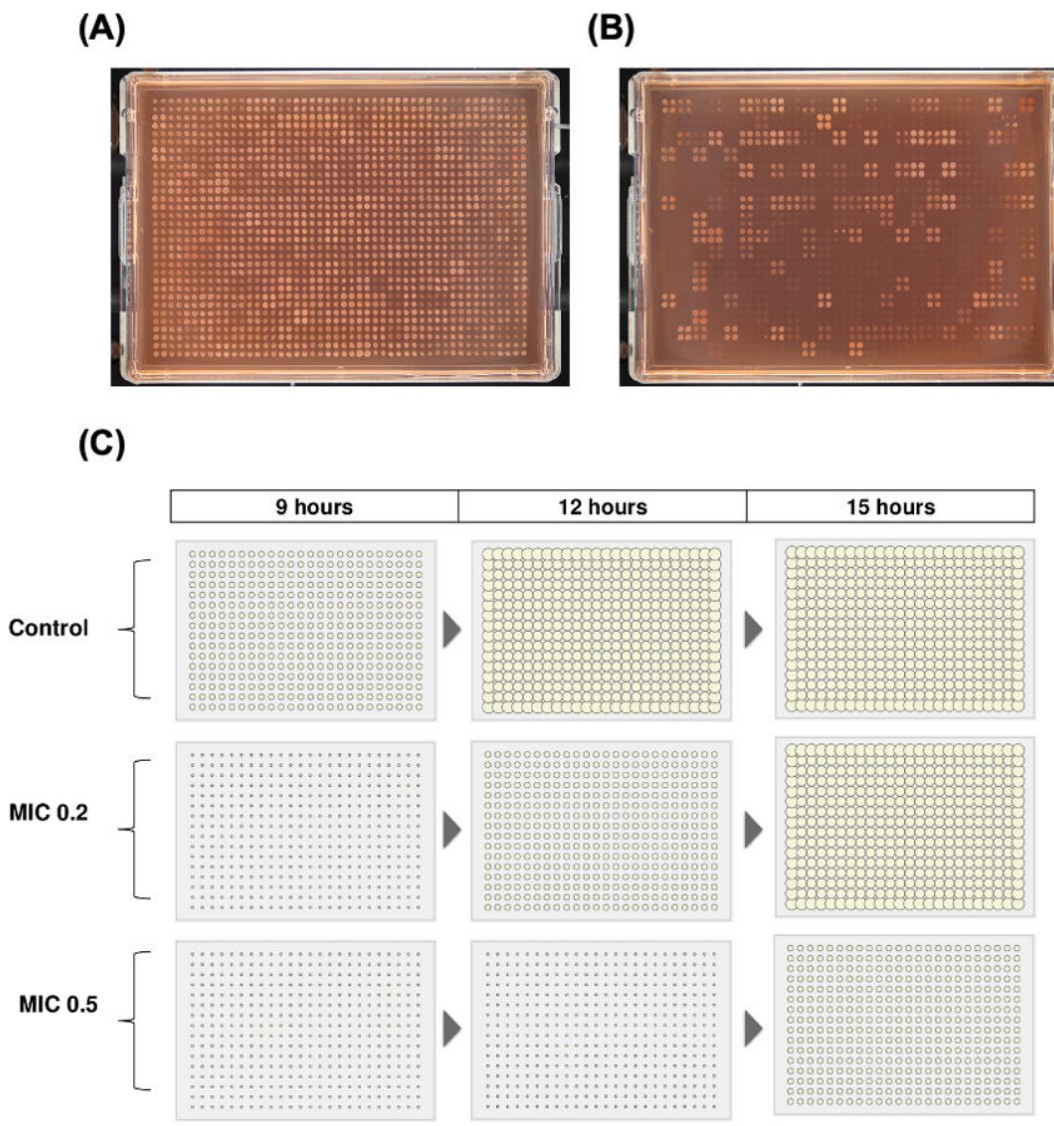

**FIG 3** Comparison of colony growth. (A) Colonies on plates containing ideal stress conditions and (B) high stress conditions. Colonies grown on plates with ideal stress conditions will have the suitability to be analyzed via iris (see "Taking images and Iris"); otherwise, if the concentrations are too high, iris analysis will not work due to the potential of low colony size and too many empty wells on the plate. (C) The relationship between incubation time and stress concentration, both of which are key determinants of colony growth. The control plate demonstrates that colony growth increases with time; however, excessive incubation can ultimately lead to overgrowth, making colonies unsuitable for precise detection using tools like Iris. At a relatively low stress of 0.2× MIC, colony size is initially inhibited, and 9 h of incubation is insufficient for colony detection. However, colony size progressively improves with extended incubation, reaching optimal size at 12 h before becoming overgrown with further incubation. In contrast, at a higher stress of 0.5× MIC, colonies reach optimal size at longer incubation times. This highlights the interplay between incubation time and stress levels relative to MIC, emphasizing how these factors should be considered when optimizing experimental conditions to ensure robust phenotypic assessments (see also: Table 3).

*Method*

1. **Capture images**: We recommend using an automated plate imaging system that is able to capture pictures of condition plates with consistent lighting and focus (Table 5).
2. **File naming and analysis**: Name image files using the format: *Condition_Concentration_SourcePlate_Replicate*. Use ("_") exclusively as separators rather than

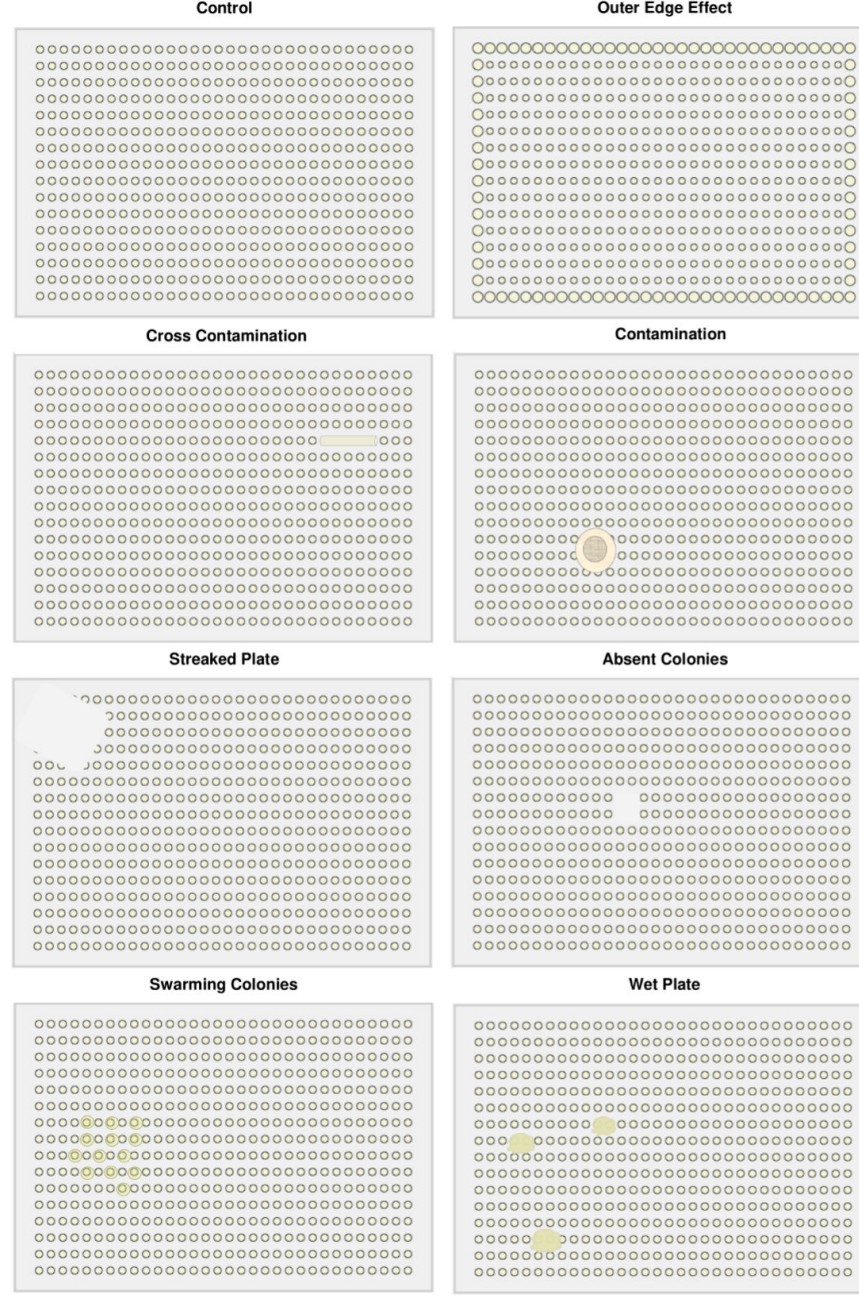

**FIG 4** Common plate effect and contamination issues encountered during chemical screening. Control plate - colonies are easily distinguishable with no observable bias; Outer edge effects - colonies at the plate's periphery grow larger compared to those in the center; Cross-contamination - bacteria from one site contaminating other locations, often forming string-like patterns due to biofilm formation; Contamination - other microbial species appear as distinct foreign colonies; Streaked plates - missing colonies due to uneven surfaces and incomplete stamping; Absent colonies - strains are entirely missing from the plate; Swarming colonies - certain strains spread actively, contaminating neighboring strains; Wet plates - inadequate drying of a plate, causing stamped strains to spread within residual water droplets affecting adjacent colony growth.

hyphens or other characters. This format is optimized for Iris but works with other analysis tools as well. For full Iris method details, see reference 1 (Table 5).

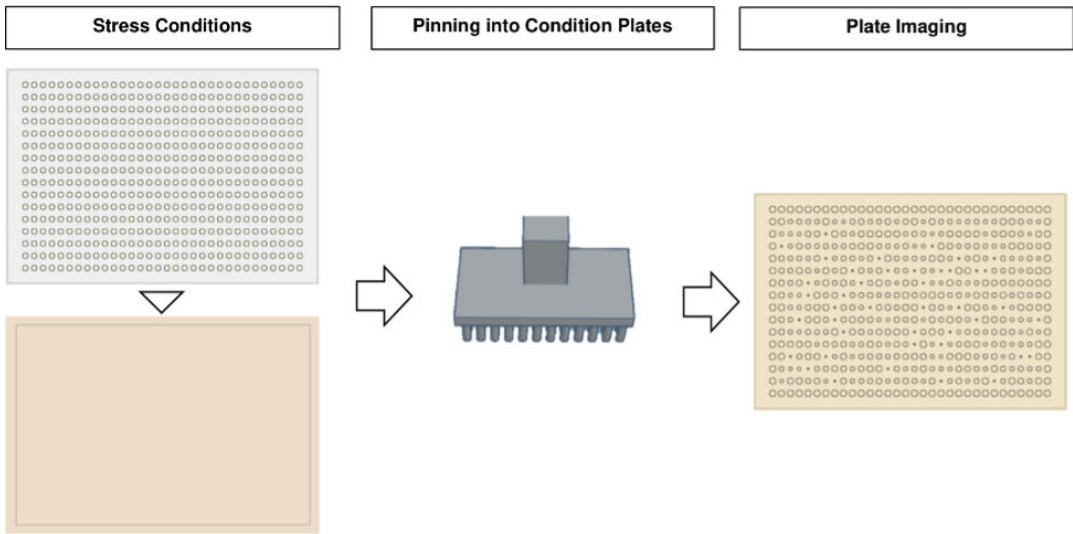

**FIG 5** Screening process. Colonies from source plates are pinned into condition plates containing the test conditions (e.g., chemical or environmental stresses) and then incubated for a selected period. The plates are then imaged to assess the study of the effects of the conditions on the mutant colonies of bacteria.

## Phenotypic analysis and visualization

This section describes the use of two complementary tools for analyzing and visualizing chemical-genomic data sets. ChemGAPP (9) processes raw colony size data and generates quantitative fitness scores, while ChemGenXplore (26) provides an interactive platform for statistical filtering, phenotype exploration, and visualization. The complete workflow is summarized in Fig. 7.

### *Phenotypic analysis (ChemGAPP)*

#### *Purpose*

This section outlines the use of ChemGAPP, an open-source platform designed to analyze microbial fitness data from chemical genomic screens (9). ChemGAPP includes two distinct pipelines tailored to different experiment scales:

1. **ChemGAPP Big:** for large-scale, genome-wide screens.
2. **ChemGAPP Small:** for targeted hypothesis testing or small-scale mutant analyses.

Users can generate fitness scores by uploading ".iris" files (from image analysis software such as Iris) into ChemGAPP, which automatically processes the data through

**TABLE 4** Screening challenges and solutions

| Challenge | Issue | Solution |
|---|---|---|
| Condensation | Plates develop condensation on the lid after bringing the plates up to room temperature. | Remove all condensation by wiping the plates with a tissue that has been pre-wet with 70% ethanol for sterility. |
| Importance of sterilization | Potential for cross-contamination when using a reusable pinning tool. | Ensure thorough sterilization between sources. Debris can be removed using a brush and water, and the pins can be sterilized by submerging the head in 70% ethanol for 10 s and allowing it to air dry. Alternative methods, such as autoclaving or using Virkon solutions, may be used depending on the tool, as well as disposable pinning heads (see "Screening design"). Refer to the manufacturers' protocols for this purpose. |
| Order of plate stamping | When stamping condition plates from source plates, the order of conditions matters. The initial stamping from a source plate transfers more cell mass, potentially affecting results. | It is recommended to assign the first condition plate containing a stress which is expected to elicit a strong effect to minimize the impact of the higher cell mass. |

**FIG 6** Process of image analysis using Iris. The imaged plates are then processed via Iris software which analyses the image and generates phenotypic scores, such as colony size, biofilm, morphology, etc. (*Note:* The "Iris Files Generation" panel depicts a representative zoomed-in section derived from the initial "imaged plates" example).

quality control, normalization, and scoring steps (Table 6). The software and documentation are available at https://github.com/HannahMDoherty/ChemGAPP.

*Materials*

Install ChemGAPP via the GitHub repository above. This tool will process Iris image data and calculate strain fitness scores.

*Method*

1. **Large-scale screens—ChemGAPP Big:** ChemGAPP Big (Fig. 7) is designed for full-genome chemical-genomic screens. The pipeline begins with optional quality control (QC) checks to flag poor-quality replicates or plates. It then performs two stages of normalization:
   a. Edge effect correction: A Wilcoxon rank-sum test compares colony sizes at the plate center vs edge to correct for border artifacts.
   b. Scaling: All plates are adjusted so their medians match the average colony size at the plate center, standardizing across conditions.

Replicate reproducibility can be evaluated to assess data quality and QC thresholds can be adjusted accordingly. After normalization, ChemGAPP Big computes S-scores, which quantify the fitness of each mutant in each condition based on the deviation from expected colony size. The scored data set can be further analyzed externally by applying a false discovery rate (FDR) threshold, commonly set at 5%, to identify statistically significant hits prior to downstream analyses, such as clustering or enrichment (26).

**TABLE 5** Imaging & Data Analysis challenges and solutions

| Challenge | Issue | Solution |
| --- | --- | --- |
| Image lighting | Avoid using flash to prevent glare from the agar. | If additional lighting is needed, position the light source to the side of the plate. |
| Quality of camera | Capturing good-quality images. | Use a high-quality camera capable of producing images with a minimum resolution of 18 megapixels. |
| Naming system | Avoiding confusion during analysis. | Establish a logical and coherent naming system for plates. |

2. **Small-scale screens—ChemGAPP Small**: ChemGAPP Small is suitable for focused experiments involving a limited number of strains or conditions. Unlike Chem-GAPP Big, it does not assume a defined fitness distribution.

Fitness is calculated either as:

a. Mutant vs wild type: using the ratio of mean colony sizes.
b. Mutant vs control condition: allowing inference of fitness under treatment vs. baseline.

Fitness scores can be visualized using heatmaps, bar plots, or swarm plots. Bar plots and heatmaps are generated by dividing the mean fitness score for each strain by the mean wild-type fitness score or under the control condition and are displayed with 95% confidence intervals. Swarm plots show the fitness ratio of each individual colony, calculated by dividing each fitness score, such as colony size, by the mean colony size of the wild type or under the control condition. Statistical significance in swarm plots is assessed using one-way analysis of variance (ANOVA).

To demonstrate the application of this workflow in practice, we include a representative data set with quality control metrics and phenotypic outputs (see "Case study: Screening a *Klebsiella pneumoniae* gene deletion library against vancomycin"). This example illustrates how ChemGAPP can be used to assess reproducibility, identify statistically significant fitness effects, and visualize strain-specific responses under chemical stress. The case study provides first-time users with a practical reference point for interpreting outputs and evaluating the robustness of their own data.

3. **To run ChemGAPP:**
   a. Export data from your plate reader or image analyzer in ".iris" format.
   b. Upload the files to ChemGAPP via the web interface or command line.

See the GitHub repository for full instructions and walkthroughs. ChemGAPP may require modification of the requirements.txt file when run on newer systems, as some package versions listed in the original configuration may be outdated. Adjusting these dependencies can help ensure compatibility during installation and execution.

### *Data visualization (ChemGenXplore)*

#### *Purpose*

This section outlines the use of ChemGenXplore, a Shiny application for interactive visualization of scored chemical-genomic data sets, including significance-based (FDR) filtering, phenotype exploration, correlation analyses, and heatmap clustering (26). The application is available online at https://chemgenxplore.kaust.edu.sa, with source code accessible via GitHub ().

#### *Materials*

CSV file containing phenotype or fitness scores (e.g., using ChemGAPP pipelines described earlier, see "Phenotypic analysis - ChemGAPP."

#### *Method*

1. **Upload:** Go to Upload Your Data set tab and upload the CSV file.
2. **Statistical Significance:** Per condition, the app computes robust z-scores (median/MAD*), converts to two-tailed *P*-values, and applies Benjamini–Hochberg FDR (27).

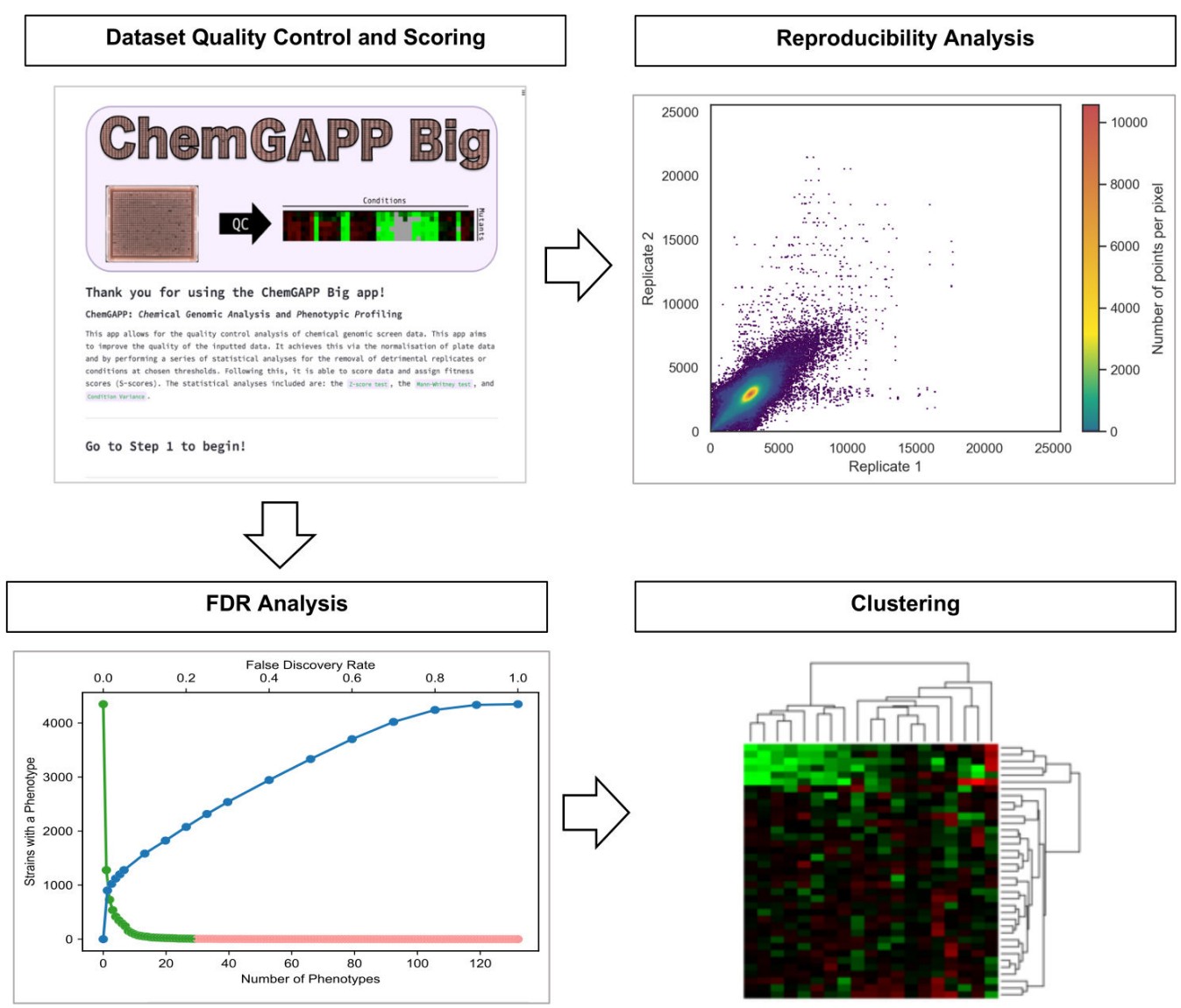

**FIG 7** Phenotypic analysis process. The ChemGAPP workflow begins with an optional quality control step to prepare the data set for scoring. Following quality control, ChemGAPP generates a reproducibility plot to assess data consistency. Subsequent analysis, performed outside of ChemGAPP, involves conducting a false discovery rate (FDR) analysis to identify statistically significant phenotypes. Clustering analysis is used to find patterns across different strains or mutants under various conditions. Data processed in ChemGAPP can subsequently be uploaded into ChemGenXplore for interactive filtering, correlation analysis, and heatmap visualization.

The app outputs a combined table of score and FDR for each gene-condition pair for further filtering.

3. **Phenotype classification and filtering:** Interactive tables and plots for all and significant phenotypes using a user-defined FDR threshold.
4. **Correlations (gene–gene and condition–condition):** Computes Pearson correlation coefficients (r) and *P*-values. Adjusts *P*-values using FDR and filters for significant correlations with |r| > 0.4.
5. **Interactive heatmaps:** Support hierarchical clustering on rows and/or columns.

**TABLE 6** Phenotypic analysis challenges and solutions

| Challenge | Issue | Solution |
|---|---|---|
| File format | ChemGAPP only accepts ".iris" files. | If using other analysis software, the output may need to be converted. |
| File naming | Proper naming is essential to avoid analysis errors. | ChemGAPP includes a file renaming tool to standardize filenames and avoid analysis errors. |

## Case study: screening a *Klebsiella pneumoniae* gene deletion library against vancomycin

To illustrate how the described workflow can be implemented in practice, we present a representative case study applying the step-by-step chemical genomic screening protocol to a single-gene deletion library of *K. pneumoniae* (28) against the bacterial cell wall targeting antibiotic vancomycin.

### Pre-screening

To find the most suitable vancomycin concentration, a representative *K. pneumoniae* library plate (384-well format) was pinned onto a fresh LB agar plate and incubated at 37°C for approximately 6 h to create a source plate for this pre-test (Fig. 8A). Next, this source plate was pinned into LB agar plates supplemented with vancomycin across a concentration range of 0–200 μg/mL (Fig. 8B). Colony growth was compared across all vancomycin concentrations and 200 μg/mL selected as it produced consistent and reproducible growth inhibition without complete arrest (Fig. 8B).

### Screening

A new *K. pneumoniae* library source plate was pinned onto LB agar plates containing 200 μg/mL vancomycin, incubated at 37°C for 12 h and subsequently imaged. We performed the same procedure simultaneously using multiple identical source plates to produce three independent biological replicates (Fig. 8C).

### Data analysis

We subsequently analyzed the imaged screening plates using the software Iris (1) that quantifies colony size by counting image pixels and produces .iris files as output for our downstream data analysis (Fig. 8D and E; Table S1). Quality control was performed by assessing reproducibility across all three biological replicates via pairwise Pearson correlation of colony sizes. Replicates showed high correlation with r = 0.889 (95% CI, 0.866–0.907; $P < 2.2 \times 10^{-16}$) (Fig. 8F), indicating that the generated raw data is highly reproducible. To quantify fitness values for each *K. pneumoniae* mutant, we used the software ChemGAPP (9) that is able to normalize raw colony-size measurements and subsequently generate fitness scores (S-scores) (Fig. 8G through I, Table S2 and Table S3 ). We next addressed multiple testing by controlling the FDR at 5% using the Benjamini–Hochberg procedure (27), and we considered phenotypes with an FDR ≤ 0.05 as statistically significant (Fig. 8H, Table S4 and Table S5 ). This FDR correction was performed in ChemGenXplore (26), and the set of genes meeting the 5% threshold was exported for downstream analyses (Table S6). Gene Ontology over-representation analysis (ORA) was performed on mutants with significant phenotypes using all assayed genes with valid GO annotations as the background set. Results are summarized in Fig. 8J. The ranked bar chart and the semantic-similarity network reveal a dominant membrane-centered signal. This is consistent with vancomycin's known envelope-targeting mode of action. Among the highest-scoring hits were *KPNIH1_07910* (colicin uptake protein TolR), *KPNIH1_07925* (peptidoglycan-associated outer membrane lipoprotein), and *KPNIH1_09025* (amino acid ABC transporter permease), all genes linked

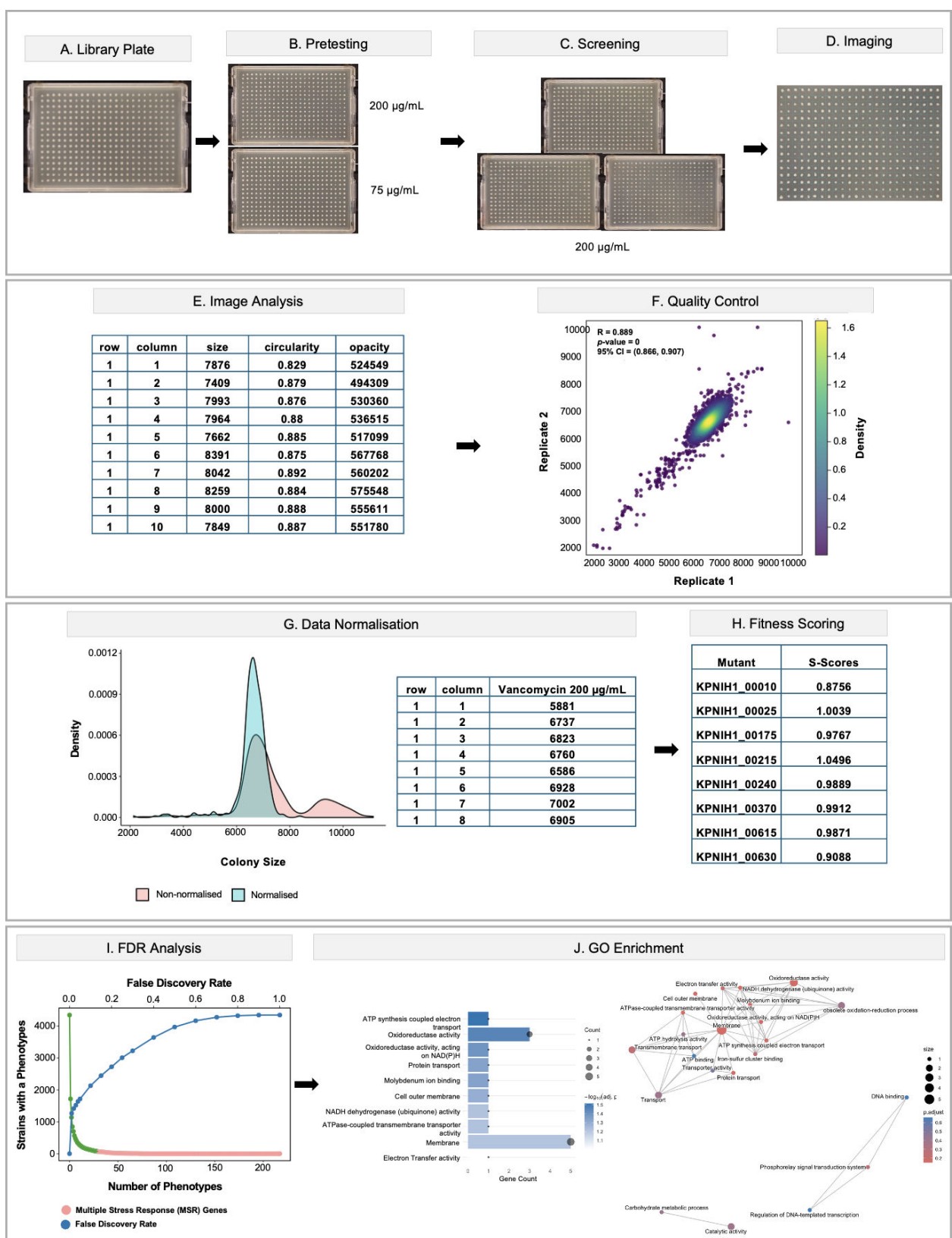

**FIG 8** Screening a *Klebsiella pneumoniae* gene deletion library against vancomycin. (A) Recovery of the deletion library on LB agar before stress exposure. (B) Pre-testing vancomycin (75–200 µg mL$^{-1}$) to select an optimal screening concentration; 200 µg mL$^{-1}$ provided measurable inhibition without complete growth arrest. (C) Triplicate screens on vancomycin plates. (D and E) Representative plate images and Iris-based colony size quantification. (F) Biological

Fig 8 (Continued)

replicates showed strong agreement (Pearson r = 0.889). (G) Data normalization performed in ChemGAPP. (H) ChemGAPP-generated S-scores quantify fitness effects. (I) Application of a Benjamini–Hochberg correction to S-scores (FDR ≤ 0.05) identified significant phenotypes. (J) Gene Ontology over-representation analysis highlighted membrane and cell-envelope functions, consistent with vancomycin's mode of action. Top hits included *KPNIH1_07910*, *KPNIH1_07925*, and *KPNIH1_09025 (see text below).*

to membrane and cell envelope functions. In summary, our case study robustly identified *K. pneumoniae* mutants exhibiting fitness defects when challenged with the antibiotic vancomycin.

## RESULTS AND DISCUSSION

### Additional considerations

#### *Troubleshooting*

When undertaking chemical genomics, there are many sources of plate effects and/or contamination issues that can be introduced (shown in Fig. 4). Many of these issues are highlighted in Table 7.

#### *Recycling*

All plates used in the screen can be sterilized and re-used, thereby cutting costs. To do this, the following method of cleaning the plates is advised:

1. **Clean used plates**: Remove old agar with a spatula, taking care not to scratch the plates. Place empty plates in an airtight box and soak for 30 min in 5% (w/v) Virkon or an appropriate laboratory disinfectant.
2. **Rinse and dry**: Rinse thoroughly with water to remove disinfectant residue. Dry plates in a 60°C incubator for at least 24 h. To reduce excess moisture, place a salt desiccant in the incubator.
3. **Inspect and store**: Discard any damaged plates. Store clean, undamaged plates in a sealed container for future use.

#### *Biofilm screening*

Biofilm formation occurs when bacterial cells encase themselves in a protective extracellular matrix (ECM) made of secreted polysaccharides and proteins, improving their survival under stress (29). This complex process is driven by multiple genes and influenced by environmental factors. Chemical genomics using single-gene knockout libraries enables efficient analysis of genes involved in biofilm production, reducing the effort of traditional assays.

For biofilm screening, the protocol can be extended to quantify biofilm formation in both solid and liquid formats. Dyes can be incorporated into the growth media that bind ECM components, such as Congo red and Coomassie blue, which have been used to stain *E. coli* and *P. aeruginosa* colonies, with increased Congo red intensity serving as a proxy for biofilm production (1, 29).

#### *Pre-testing biofilm conditions*

Begin pre-testing with Congo red alone. If mutant visibility or Iris quantification is poor, add Coomassie blue in varying ratios. An effective starting ratio, as seen in *P. aeruginosa*, is 2:1 Congo red to Coomassie blue (e.g., 40:20 µg/mL). Once optimized, plates should show clear background and distinct coloration—red-stained mutants indicate biofilm production, while white mutants are non-producers. In species with strong biofilm formation, dye use enhances mutant detection, as shown in Fig. 9: (A) no dye, (B) with dye, (C) enhanced biofilm visibility.

**TABLE 7** Troubleshooting

| Issue | Possible reason | Solution |
|---|---|---|
| Uneven shading across the plate. | Insufficient mixing of LB media before plate pouring. | Ensure thorough mixing of agar. It is recommended to use a magnetic stirrer. |
| Unstable agar under certain conditions. | Insufficient mixing of the condition and/or the condition is not stable within the media. | Use a magnetic stirrer and/or check and optimize the suitability of condition concentration, such as adjusting the pH to ensure agar solidifies. |
| | Conditions can prevent the agar from solidifying, such as pH and sodium dodecyl sulfate (SDS). | |
| Bubbles in the agar. | Some detergents, such as SDS, have a greater tendency to form bubbles within the agar. | A flaming technique can be used to resolve this issue. |
| Fungal contamination after incubation. | High humidity in the incubator. | Use humidity-controlled incubators (max 20% humidity). Place a water tray in incubators with low humidity. Consider the use of an antifungal to decontaminate. |
| Outer edge effects of colonies and distortion. | Increased growth at the edge of solid agar due to reduced nutrient competition. This problem is exacerbated if present on source plates. | Ensure adequate mixing of media and chemicals/dyes before plate pouring. Reduce incubation time. |
| Biofilm formation (sticky/thin glossy covering on plate). | Plates are left too long in the incubator. | Reduce incubation time and/or temperature. |
| | Species are prone to biofilm formation. | Use broth media with higher salt content and avoid no salt conditions. |
| | Unsuitable conditions can enhance biofilm formation. | |
| | Plates are too cold, affecting colony attachment. | Allow plates to adjust fully to room temperature before use. |
| Colonies are fusing with each other. | Plates are overgrown plates leading to cross-contamination. | Reduce incubation time and/or temperature. Store plates in the fridge after incubation. Ensure agar is uniformly distributed. |
| | Incubation time is too long. | |
| No colonies observed on plates. | Poor transfer of bacterial strains or poor pre-testing conditions. | Check library plates for growth on normal media. Verify pre-testing conditions or perform broth microdilution assays to obtain MIC values. |
| Reduced colony size. | Plates have not been incubated long enough. | Incubate the plate for a longer time. |
| | Condition concentrations may be too high. | Reduce the condition plate concentrations. |
| Colonies show swarming effects/motility. | Media is not ideal for bacteria or plates incubated for too long. | Alter the salt composition of media, shorten incubation time, or use a smaller plate format. |

In addition to the use of Congo red and Coomassie blue on solid media for visual assessment (described previously), microtiter plate-based assays can be used for liquid biofilm formation. This involves using sterile 96-well polystyrene flat-bottomed plates that promote surface attachment and biofilm formation (tissue-culture-treated plates can also be used to further enhance adherence). In liquid cultures, biofilm quantification is commonly quantified using crystal violet staining. This process usually involves growing bacteria in microtiter plates, staining the biofilm with crystal violet, washing away excess dye, and then solubilizing the bound crystal violet for spectrophotometric quantification. The plates are then incubated for a set period of time depending on the species. Stress conditions are added either at the start or after initial biofilm formation to analyze biofilm resistance or eradication (1,29).

## Species-specific considerations

Below is a non-exhaustive list of the media requirements for various bacterial species. Note that in all options listed, the type/amount of agar remains the same.

- *E. coli*, *Vibrio cholerae*, *P. aeruginosa,* and *K. pneumoniae* can be grown on agar made by combining LB Lennox Broth (Tryptone 10 g/L, NaCl 5 g/L, yeast extract 5 g/L) and Agar 20 g/L.
- *S. enterica* on LB Miller Broth (Tryptone 10 g/L, NaCl 10 g/L, Yeast Extract 5 g/L) and Agar 15 g/L.

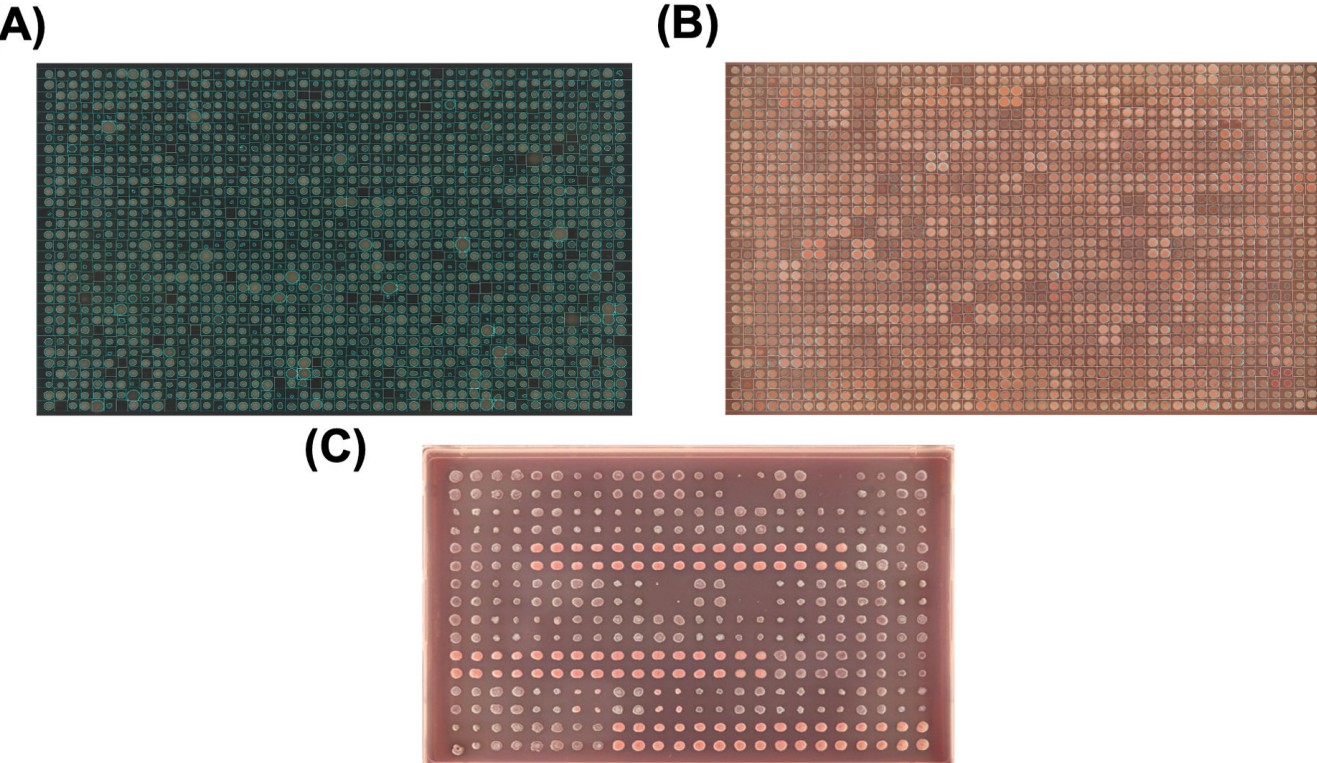

**FIG 9** Impact of bacterial colony detection when grown on unstained vs stained plates for Iris quantification. (A) Unstained plates with limited Iris colony detection. (B) Stained plates (2:1 Congo red:Coomassie blue) improve Iris colony detection via enhanced contrast. (C) Visual identification of biofilm formation among mutants is improved by staining, even without software quantification.

- *Mycobacterium bovis* Bacillus Calmette-Guérin (BCG) and *Mycobacteroides abscessus* can be grown using an enriched 7H9 media supplemented with 1% (w/v) glycerol, 10% (w/v) ADC, 0.2% (w/v) Bacto cas amino acids, 50 µg/mL L-Tryptophan, and 50 µg/mL Natures Aid multivitamins and then plated onto 7H9 agar using 2% (w/v) BD Difco agar and 7H9 media. Optionally, add 25 µg/mL of amphotericin B to inhibit fungal contamination. To prevent drying out during the incubation period, ensure to monitor the humidity using a hygrometer if possible or by using a water reservoir and minimizing airflow. The liquid BCG culture that is being pinned should be shaken at 25 rpm before use in the pinning operation.

When performing chemical screening using bacteria, it is crucial to consider how different bacterial species appear on the plates, as the morphology of bacterial colonies can provide valuable information about growth patterns, behavior, and potential antimicrobial effects. The appearance of bacterial colonies can vary significantly based on species-specific characteristics, such as colony shape, color, size, texture, and growth pattern, all of which can influence the interpretation of chemical screening results. Certain species (e.g., *P. aeruginosa*, *E. coli*, or *S. enterica*) may have either intrinsic or acquired resistance to multiple antibiotics. Therefore, it's important to consider the strain's resistance profile when interpreting the results of chemical screening for antimicrobial activity.

Our research group has undertaken screening analysis on several species as listed in this section. Highlighted below in Table 8 are species-specific observations taken from the experiences of our group.

## Screening design

### Screen format

Screens are typically run in 96, 384, or 1,536-well formats. Higher-density formats reduce plate use, saving time and cost. However, if swarming strains are present, they may spread into adjacent colonies. In such cases, a lower-density format may be needed to provide more space and reduce interference.

### Pinning platform

As highlighted in "Equipment Specificity," this protocol uses the S&P Robotics BM series platform, which features a reusable pinning head, plate stacker, wash station, and UV sterilization to prevent cross-contamination. Alternative robotic platforms, such as the Singer ROTOR HDA, which uses disposable plastic pinning heads, may require adjustment of steps, including pinning procedures, sterilization methods, and plate formats. Users should consult their equipment manuals to ensure compatibility and optimal performance. For low-throughput screens, manual pinning is also an option.

### Robot cleaning

Ensure that the robot is cleaned thoroughly before commencing any screen to avoid introducing contamination and/or inconsistencies in data generation. Robot cleaning is best achieved using both 70% ethanol and in-built features of the robot such as UV sterilization and water/ethanol reservoirs. Check the robot manufacturers' guidelines for further details.

## Adapting the protocol for liquid culture screening

Although this protocol primarily focuses on solid-media based screening, it can be adapted for use in liquid culture formats which may be preferable for specific bacterial species or phenotypic assays (e.g., metabolic activity, growth kinetics, or fluorescence-based).

Below are the key modifications for liquid culture screening:

**TABLE 8** Species-Specific growth characteristics and considerations[a]

| Species | Biofilm Formation | Growth time | Motility (swarming effect) | Consistent growth on solid agar | Storage stability of source plates | Plate replication | No. condition plates from source plate |
|---|---|---|---|---|---|---|---|
| *Escherichia coli* | * | ~6 h | * | + | ** | S-S | ~11 |
| *Vibrio cholerae*[b] | ** | ~24 h | * | + | * | S-S | ~11 |
| *Pseudomonas aeruginosa* | *** | ~6 h | ** | + | *** | S-S | TBD |
| *Staphylococcus aureus* | TBD | ~16 h | * | + | *** | S-S | ~10 |
| *Salmonella enterica* | * | ~6 h | *** | + | *** | S-S | ~11 |
| *Mycobacterium bovis* BCG[c] | * | ~2 weeks | * | − | ** | L-S | TBD |
| *Acinetobacter baumannii* | **** | **** | * | + | ** | S-S | TBD |
| *Klebsiella pneumoniae* | * | ~6 h | * | − | *** | S-S | ~22 |
| | | | | | | L-S | |

[a]This table summarizes key growth characteristics of various bacterial species based on our group's experience. It includes optimal growth temperature, biofilm formation tendency, growth duration, swarming motility, and storage stability of source plates. Symbols (*) indicate relative levels, with more asterisks representing a higher degree of the characteristic (e.g., stronger biofilm formation or longer growth time). The + and − symbols denote the presence or absence of a specific trait, respectively. S indicates solid and L liquid from source plate to condition plate, i.e., S-L represents solid source plate to liquid condition plate replication. TBD—to be determined.
[b]When imaging *Vibrio cholerae*, it may be difficult to visualize the colonies when grown on agar alone. It is recommended to use the biofilm assay procedure (see "Biofilm screening") to improve colony image acquisition.
[c]The BCG is a challenging species to work with due to its low colony replication and high doubling time. Colony replication is only possible from liquid to solid growth medium which when grown at 37°C incubation time takes approximately 9 days to grow to confluency. As a result of these long growing times, there is potential for contamination to occur. One possible solution to this is to wrap the plates in parafilm during the static incubation period.

Plate type: Use sterile, flat-bottom 96-well or 384-well polystyrene microtiter plates for liquid culture assays. These plates are compatible with standard plate readers for optical density (OD) or fluorescence measurements.

Detection methods: Growth and fitness of bacterial cultures are usually assessed by measuring OD600 at specific time points using a microplate reader. Additional phenotypes can be measured using fluorescence, luminescence, or colorimetric assays, depending on the experimental design.

Inoculation and culture conditions: Inoculate wells with a standardized volume of pre-cultured strains from library plates or liquid source plates. A typical volume is 100–200 µL per well for a 96-well plate and 30 - 40 µL for a 384-well plate. Incubate plates under shaking conditions (e.g., 200–250 rpm) at a suitable temperature (e.g., 37°C) to ensure aeration and uniform growth.

Time points: Kinetic measurements over time (e.g., every 5 min to 1 h) can be taken to monitor growth curves and determine lag phase, exponential growth rate, and stationary phase onset. Alternatively, endpoint measurements can be taken at a predefined incubation duration.

Considerations for automation: Automated liquid handling systems can be used to inoculate and replicate library strains into microtiter plates. Ensure proper calibration to avoid cross-contamination and ensure consistent volume transfer.

These approaches are particularly useful for high-throughput screening of antibiotics, metabolic inhibitors, or stress responses in a more dynamic and quantitative manner than solid media screening. For detailed liquid screening protocols, reference 22 provides an excellent example of how to design and execute large-scale, high-throughput liquid culture screens for drug–drug interactions in bacteria.

## Conclusion

The potential applications of chemical genomics screens extend beyond basic microbiology research into drug discovery, environmental microbiology, and synthetic biology. As new bacterial species and stress conditions are incorporated into screening workflows, the insights gained will continue to expand our understanding of microbial function and adaptation. By following this structured protocol, researchers can generate high-quality, reproducible data sets that contribute to the broader goal of functional genomics and microbial systems biology.

To conclude, this protocol provides a comprehensive framework for performing high-throughput chemical genomic screens to uncover gene functions in bacterial strains under varying stress conditions. If you have additional questions, do not hesitate to contact us.

### ACKNOWLEDGMENTS

This work was funded by the BBSRC-funded Doctoral Training Centre. We thank members of the Banzhaf and Moradigaravand laboratories for thoughtful discussions and feedback.

This work was supported by the Biotechnology and Biological Sciences Research Council (BBSRC) and University of Birmingham funded Midlands Integrative Biosciences Training Partnership (MIBTP) (BB/T00746X/1). In addition, support was received from the Darwin Trust of Edinburgh Charity. This work was supported by a UKRI Future Leaders Fellowship (MR/V027204/1) to Manuel Banzhaf. KAUST authors were supported by KAUST faculty baseline fund (BAS/1/1108-01-01). They were also supported by FCC/1/5932-01-03 from KAUST Center of Excellence for Smart Health.

M.B., G.W., M.T.M., H.A., and S.S. wrote and edited the manuscript. G.W., M.B., M.T.M., and D.M. conceptualized the project. J.H., S.B., A.J.H., H.M.D., R.S., M.A., X.M., Q.X., J.B., M.G., P.B., and P.M., contributed resources and contributed towards the reviewing process. All authors have read and approved the manuscript.

## AUTHOR AFFILIATIONS

[1]Institute of Microbiology and Infection, University of Birmingham, Birmingham, United Kingdom

[2]Laboratory of Infectious Disease Epidemiology, KAUST Center of Excellence for Smart Health and Biological and Environmental Science and Engineering (BESE) Division, King Abdullah University of Science and Technology (KAUST), Thuwal, Kingdom of Saudi Arabia

[3]Biosciences Institute, Faculty of Medical Sciences, Newcastle University, Newcastle upon Tyne, United Kingdom

[4]School of Pharmacy, University of Nottingham, Nottingham, United Kingdom

[5]Department of Bacterial Molecular Genetics, Faculty of Biology, University of Gdansk, Gdansk, Poland

## AUTHOR ORCIDs

Georgia Williams http://orcid.org/0009-0003-0533-0383
Mathew T. Milner http://orcid.org/0009-0004-8653-4640
Danesh Moradigaravand http://orcid.org/0000-0001-6652-5617
Manuel Banzhaf http://orcid.org/0000-0002-4682-1037

## FUNDING

| Funder | Grant(s) | Author(s) |
| --- | --- | --- |
| Biotechnology and Biological Sciences Research Council | BB/T00746X/1 | Georgia Williams |
| Darwin Trust of Edinburgh | | Huda Ahmad |
| Biotechnology and Biological Sciences Research Council | | Susan Sutherland |
| UK Research and Innovation | MR/V027204/1 | Manuel Banzhaf |
| King Abdullah University of Science and Technology | BAS/1/1108-01-01 | Danesh Moradigaravand |
| King Abdullah University of Science and Technology | FCC/1/5932-01-03 | Danesh Moradigaravand |

## AUTHOR CONTRIBUTIONS

Georgia Williams, Conceptualization, Data curation, Formal analysis, Investigation, Methodology, Validation, Visualization, Writing – original draft, Writing – review and editing | Huda Ahmad, Conceptualization, Data curation, Formal analysis, Investigation, Methodology, Software, Validation, Visualization, Writing – original draft, Writing – review and editing | Susan Sutherland, Conceptualization, Data curation, Formal analysis, Investigation, Methodology, Validation, Visualization, Writing – original draft, Writing – review and editing | James Haycocks, Conceptualization, Data curation, Investigation, Methodology, Visualization, Writing – review and editing | Sam Benedict, Conceptualization, Data curation, Investigation, Methodology, Visualization, Writing – review and editing | Adam J. Hart, Conceptualization, Data curation, Investigation, Methodology, Visualization, Writing – review and editing | Hannah H. Doherty, Methodology, Software | Rudi Sullivan, Resources | Micheal Alao, Methodology | Xuyu Ma, Writing – review and editing | Qianhui Xu, Writing – review and editing | Jack Bryant, Writing – review and editing | Monika Glinkowska, Resources, Writing – review and editing | Peter Banks, Writing – review and editing | Patrick Moynihan, Funding acquisition, Resources, Writing – review and editing | Mathew T. Milner, Conceptualization, Data curation, Formal analysis, Investigation, Methodology, Software, Validation, Visualization, Writing – original draft, Writing – review and editing | Danesh Moradigaravand, Conceptualization, Data curation, Formal analysis, Funding acquisition, Investigation, Methodology,

Software, Validation, Visualization, Writing – review and editing | Manuel Banzhaf, Conceptualization, Data curation, Formal analysis, Funding acquisition, Investigation, Methodology, Project administration, Resources, Software, Supervision, Validation, Visualization, Writing – original draft, Writing – review and editing

## DATA AVAILABILITY

Raw data from the case study are provided in the supplementary files. The analysis software used in this workflow (Iris, ChemGAPP & ChemGenXplore) is available at (v0.9.7, https://github.com/critichu/Iris), https://github.com/HannahMDoherty/ChemGAPP, and (https://github.com/Hudaahmadd/ChemGenXplore).

## ADDITIONAL FILES

The following material is available online.

### Supplemental Material

**Table S1 (mSystems00885-25-S0001.csv).** Raw fitness measurements from pre-testing plates.
**Table S2 (mSystems00885-25-S0002.csv).** Initial data set after ChemGAPP normalization.
**Table S3 (mSystems00885-25-S0003.csv).** Scored data set from ChemGAPP for pretested conditions.
**Table S4 (mSystems00885-25-S0004.txt).** Scored data set containing Benjamini-Hochberg adjusted *P*-values.
**Table S5 (mSystems00885-25-S0005.xlsx).** Full scored data set from screening vancomycin.
**Table S6 (mSystems00885-25-S0006.xlsx).** Subset of the vancomycin data set including only mutants with Benjamini-Hochberg adjusted *P*-values.

### Open Peer Review

**PEER REVIEW HISTORY (review-history.pdf).** An accounting of the reviewer comments and feedback.

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
