## [Reviewer comments · mSystems]

High-throughput chemical genomic screening: a step-by-step workflow from plate to phenotype

Georgia Williams, Huda Ahmad, Susan Sutherland, James Haycocks, Samuel Benedict, Adam Hart, Rudi Sullivan, Xuyu Ma, Qianhui Xu, Jack Bryant, Monika Glinkowska, Peter Banks, Patrick Moynihan, Mathew Milner, Danesh Moradigaravand, Manuel Banzhaf, Hannah Doherty, and Micheal Alao

Corresponding Author(s): Manuel Banzhaf, Newcastle University

Review Timeline:

Submission Date:	June 23, 2025
Editorial Decision:	July 30, 2025
Revision Received:	October 6, 2025
Editorial Decision:	October 15, 2025
Revision Received:	October 16, 2025
Accepted:	October 21, 2025

Editor: Julia Willett

Reviewer(s): Disclosure of reviewer identity is with reference to reviewer comments included in decision letter(s). The following individuals involved in review of your submission have agreed to reveal their identity: Xue Liu (Reviewer #1)

Transaction Report:

DOI: <https://doi.org/10.1128/msystems.00885-25>

Re: mSystems00885-25 (**High-throughput chemical genomic screening: a step-by-step workflow from plate to phenotype**)

Dear Dr. Manuel Banzhaf:

As you can see, this manuscript has been carefully reviewed by 2 expert scientists. Both reviews highlight the strengths of the work and suggest changes that I agree would strengthen the manuscript. Please revise according to these points, including the attachment with comments from reviewer 2. Another point of consideration to address is how these screens would need to be adapted for biosafety containment if the organism is pathogenic.

Revision Guidelines

Sincerely,
Julia Willett
Editor
mSystems

Reviewer #1 (Comments for the Author):

This manuscript presents a detailed, step-by-step protocol for conducting high-throughput chemical genomic screens in bacteria,

describing each experimental and analytical step from plate preparation through data handling and quality control. The authors provide practical guidance, troubleshooting tips, and discuss software tools for image analysis and result visualization. The protocol is intended to enable robust and reproducible screening of bacterial phenotypes in response to chemical perturbations, making it accessible to both new and experienced researchers in microbial genetics and chemical biology.

Overall, the manuscript is clearly written, thoughtfully organized, and highly practical. The comprehensive coverage of the workflow, attention to experimental details, and focus on reproducibility and troubleshooting are particularly commendable. The stepwise structure of the protocol, along with integration of relevant software pipelines, adds significant practical value to the community.

However, several points and suggestions are offered below to further improve the clarity, applicability, and broader impact of the work.

1) When transferring cells from library/source plates to screening plates, physiological variability between bacterial cultures could lead to artefactual results. It would strengthen the protocol to include recommendations on how to minimize this, such as:

Synchronization of growth phase prior to spotting,

Consistent pre-culture conditions,

Use of internal plate controls or interleaved replicates,

Suggested metrics for assessing spotting consistency.

2) Including a real-world application of the protocol - ideally with representative data, QC metrics, and phenotypic outputs - would greatly enhance accessibility for first-time users. This could be a brief case study embedded within the manuscript or a supplementary dataset linked to the methods.

3) While the protocol briefly mentions liquid culture and biofilm assays, this section could be expanded or linked more explicitly to existing protocols. Even a short paragraph highlighting necessary modifications (e.g., plate types, detection methods, time points) would broaden the scope and applicability.

4) A brief comparison in the introduction part between arrayed library-based screening and sequencing-based pooled screening strategies (e.g., Tn-seq, CRISPRi-seq) would help contextualize the advantages and limitations of this approach. This could include discussion of:
Throughput vs. resolution, Cost, Scalability, Suitability for essential gene screening or strain-specific responses.

Section 1.1: The temperature unit should be formatted correctly as "5565 {degree sign}C" using an proper degree symbol. VWR one-well plates: It would be helpful to specify the dimensions (e.g., diameter) of the plates for reproducibility. *Mycobacterium bovis* BCG should be italicized.

Line 475: Replace "Bacto™" with "Bacto{trade mark, serif}" to correct the formatting of the trademark symbol.

This is an excellent and highly useful protocol that will benefit many researchers in microbiology, chemical biology, and functional genomics. With attention to the points above - particularly the broader applicability, clarification of experimental controls, and inclusion of representative data - this protocol could serve as a robust reference for bacterial phenotype screening.

Reviewer #2 (Comments for the Author):

In general the manuscript is of high interest as a starting point for any scientist who wants to perform a chemical genomics screen specially using Bacteria. It is well explained and the troubleshooting tables would be of help for any beginner in high-throughput screens.

Review

In general the manuscript is of high interest as a starting point for any scientist who wants to perform a chemical genomics screen in Bacteria. The troubleshooting tables would be of help for any beginner in high-throughput screens.

Comments to improve the rigour and clarity (line by line):

General- Please clarify early in the manuscript that this protocol is tailored specifically for the pinning robot used in the authors' laboratory. Several steps may vary when using other robotic platforms, such as the Singer ROTOR. Acknowledging this explicitly will help readers adapt the protocol appropriately to their own equipment.

Line 54 – References: the most recent reference from the biggest chemical genomics study in *S. cerevisiae* is missing, cite: Vieitez C, et al. Nature Biotech 2022 <https://pubmed.ncbi.nlm.nih.gov/34663920/> When mentioning Tong et al 2001 and Schuldiner et al 2005 mention that are high throughput screens but NOT chemical genomics.

Line 116- when talking about stresses, would be more accurate talking about conditions or clarify it further to avoid misunderstanding from the reader. For example heat stress is not “added” to the plate.

Figure 2- wrong well in image (not A1, but H1)

Line 121- Add a minimum number of hours need it until turning the plates upside down

Figure 1- the image shows a source plate with 384 colonies, from the experience of this reviewer it would be better to replace the image for one with a couple of empty wells, since these empty wells are usually very informative as “contamination controls”.

Figure 3B- “otherwise if the concentrations are too high Iris analysis will not work” – clarify in the text why Iris detection won’t work (minimum colony size? Too many empty wells?, any other issue?)

Line 232- “plates are incubated based on the organism and stress condition” – link this text to Table 2 which provides further information.

Pretesting Challenges and Tips- Section 1.3 – “Consult literature data for conditions, examples such as ...” State the organism of each example to guide the reader to their favourite organism dataset.

Line 307- Please clarify that the mention of non-dispensable pinning pads is specific to the particular robot model used in this protocol, as many laboratories use dispensable pinning pads in chemical genomics workflows.

Figure 6 – Consider increasing the clarity of this figure, if the starting point is a high density plate it is confusing why the output are few colonies.

Line 349 - make sure the format is correct

Condition_Concentration_SourcePlate_Replicate (always “_” or interchange with “–”“?)

Line 379- “An alternative tool for processing growth curve data is the Kinetic Data Companion in R” this sentence is out of context since growth data was not mentioned in this protocol. I suggest to either put the sentence in context or just remove it.

Figure 8B- In the printed version of the manuscript, it is not visible that 8B shows stained colonies.

Response to Reviewers

Dear Julia Willett,

We would like to thank you and the reviewers for the constructive feedback and valuable suggestions provided on our manuscript. We have considered all comments and revised the manuscript accordingly. We believe that these revisions have increased the quality and clarity of our work.

We provide a detailed, point-by-point response to the reviewers' comments below.

For ease of review:

- **Reviewer comments** are shown in black *italicised* text.
- **Our responses** follow each comment in red text.
- Where changes/additions were made, we indicate the revised section(s) and line numbers in the manuscript.

Reviewer 1

1) When transferring cells from library/source plates to screening plates, physiological variability between bacterial cultures could lead to artefactual results. It would strengthen the protocol to include recommendations on how to minimize this, such as:

Synchronization of growth phase prior to spotting,

Consistent pre-culture conditions,

Use of internal plate controls or interleaved replicates,

Suggested metrics for assessing spotting consistency.

This point is acknowledged and a section related to address this has been incorporated into the manuscript on lines 241- 274.

2) Including a real-world application of the protocol - ideally with representative data, QC metrics, and phenotypic outputs - would greatly enhance accessibility for first-time users. This could be a brief case study embedded within the manuscript or a supplementary dataset linked to the methods.

An additional section has been added to the manuscript entitled '4. Case Study: Screening a *Klebsiella pneumoniae* Gene Deletion Library Against Vancomycin' for a real-world application of the protocol. It has been added in lines 552-611. For this case study we visualised our findings using ChemGenXplore, a Shiny application for interactive visualisation of scored chemical-genomic datasets recently developed by our laboratory. It is available as a preprint and currently under review in the journal bioinformatics. As ChemGenXplore complements ChemGAPP (section 3.2) we decided to also add a small subsection (3.2.2) to describe its usage to the reader.

We believe highlighting a web-based visualisation tool offers practical value to the reader, especially for those without prior knowledge in bioinformatics.

3) While the protocol briefly mentions liquid culture and biofilm assays, this section could be expanded or linked more explicitly to existing protocols. Even a short paragraph highlighting necessary modifications (e.g., plate types, detection methods, time points) would broaden the scope and applicability.

We acknowledge this point and have now addressed this from lines 661-669 of the manuscript as well as introducing a new section, 5.6 Adapting the Protocol for Liquid Culture Screening to cover this. We also cite the Brochado *et al.* 2018 paper for readers to find a good example of how to perform high-throughput liquid based culture based screens.

4) A brief comparison in the introduction part between arrayed library-based screening and sequencing-based pooled screening strategies (e.g., Tn-seq, CRISPRi-seq) would help contextualize the advantages and limitations of this approach. This could include discussion of: Throughput vs. resolution, Cost, Scalability, Suitability for essential gene screening or strain-specific responses.

This comparative introduction has been included in the main introduction of the manuscript from lines 48-64.

Section 1.1: The temperature unit should be formatted correctly as "5565 {degree sign}C" using an proper degree symbol. VWR one-well plates: It would be helpful to specify the dimensions (e.g., diameter) of the plates for reproducibility. *Mycobacterium bovis* BCG should be italicized.

Degree symbols have been amended and dimensions for the VWR one-well plates have been added. *Mycobacterium bovis* has been italicised.

Line 475: Replace "BactoTM" with "Bacto{trade mark, serif}" to correct the formatting of the trademark symbol.

Corrected.

Reviewer 2

General- Please clarify early in the manuscript that this protocol is tailored specifically for the pinning robot used in the authors' laboratory. Several steps may vary when using other robotic platforms, such as the Singer ROTOR. Acknowledging this explicitly will help readers adapt the protocol appropriately to their own equipment.

This clarification has been incorporated into manuscript from lines 118-126.

Line 54 – References: the most recent reference from the biggest chemical genomics study in *S. cerevisiae* is missing, cite: Vieitez C, *et al.* Nature Biotech

2022 <https://pubmed.ncbi.nlm.nih.gov/34663920/> When mentioning Tong et al 2001 and Schuldiner et al 2005 mention that are high throughput screens but NOT chemical genomics.

This reference has been added and cited in the manuscript on line 76.

Line 116- when talking about stresses, would be more accurate talking about conditions or clarify it further to avoid misunderstanding from the reader. For example heat stress is not “added” to the plate.

Re-edited the manuscript throughout so that stresses are now referred to as “conditions” to avoid misunderstanding.

Figure 2- wrong well in image (not A1, but H1)

This has been corrected.

Line 121- Add a minimum number of hours need it until turning the plates upside down

This has been amended and the number of hours specified.

Figure 1- the image shows a source plate with 384 colonies, from the experience of this reviewer it would be better to replace the image for one with a couple of empty wells, since these empty wells are usually very informative as “contamination controls”.

This image has now been replaced to show the empty wells.

Figure 3B- “otherwise if the concentrations are too high Iris analysis will not work” – clarify in the text why Iris detection won’t work (minimum colony size? Too many empty wells?, any other issue?)

This section has been clarified further on lines 292-293.

Line 232- “plates are incubated based on the organism and stress condition” – link this text to Table 2 which provides further information.

This has been linked to Table 2.

Pretesting Challenges and Tips- Section 1.3 – “Consult literature data for conditions, examples such as ...” State the organism of each example to guide the reader to their favourite organism dataset.

This suggestion has been included alongside literature citations.

Line 307- Please clarify that the mention of non-dispensable pinning pads is specific to the particular robot model used in this protocol, as many laboratories use dispensable pinning pads in chemical genomics workflows.

This comment has now been addressed from lines 391-393.

Figure 6 – Consider increasing the clarity of this figure, if the starting point is a high density plate it is confusing why the output are few colonies.

A one line description of the figure addresses the reviewers' comments related to "why the output are few colonies".

*Line 349 - make sure the format is correct
Condition_Concentration_SourcePlate_Replicate (always “_” or interchange with “–”“?)*

The format has been re-checked to ensure it is correct.

Line 379- “An alternative tool for processing growth curve data is the Kinetic Data Companion in R” this sentence is out of context since growth data was not mentioned in this protocol. I suggest to either put the sentence in context or just remove it.

The sentence has been removed for clarity.

Figure 8B- In the printed version of the manuscript, it is not visible that 8B shows stained colonies.

A clearer plate has been added that shows stained colonies.

We would like to thank you and the reviewers again for your constructive feedback. We hope the revised manuscript now meets the journal's standards and look forward to your consideration.

Yours sincerely,

Manuel Banzhaf and Georgia Willams
On behalf of all co-authors
Newcastle University

Re: mSystems00885-25R1 (**High-throughput chemical genomic screening: a step-by-step workflow from plate to phenotype**)

Dear Dr. Manuel Banzhaf:

Reviewer 2 points out a small modification that should be made to the labels in Figure 2. Please address this and resubmit.

Revision Guidelines

Sincerely,
Julia Willett
Editor
mSystems

Reviewer #1 (Comments for the Author):

The authors have addressed all my previous concerns thoroughly. I would like to thank them for providing such a clear and detailed protocol. This well-designed workflow will undoubtedly serve as a valuable reference for the field and facilitate future high-throughput chemical genomic screenings.
A couple of minor editorial points for consistency:

Line 622: *Klebsiella pneumoniae* should be italicized.

Line 93: *S. cerevisiae* should be italicized.

Reviewer #2 (Comments for the Author):

This reviewer thanks the authors for implementing all the changes and clarifications in the manuscript. I have only detected Figure 2 has not been updated yet (H1 vs A1), please update this figure before the final publication.

Response to Authors

I would like to thank the authors for addressing all comments, which help to clarify the mentioned aspects in the manuscript.

Only one Figure still needs to be corrected (Figure 2)

Response to Reviewers

Dear Julia Willett,

We would like to thank you and the reviewers for the constructive feedback and valuable suggestions provided on our manuscript. We have considered all comments and revised the manuscript accordingly. We believe that these revisions have increased the quality and clarity of our work.

We provide a detailed, point-by-point response to the reviewers' comments below.

For ease of review:

- **Reviewer comments** are shown in black *italicised* text.
- **Our responses** follow each comment in red text.
- Where changes/additions were made, we indicate the revised section(s) and line numbers in the manuscript.

Reviewer 1

1) When transferring cells from library/source plates to screening plates, physiological variability between bacterial cultures could lead to artefactual results. It would strengthen the protocol to include recommendations on how to minimize this, such as:

Synchronization of growth phase prior to spotting,

Consistent pre-culture conditions,

Use of internal plate controls or interleaved replicates,

Suggested metrics for assessing spotting consistency.

This point is acknowledged and a section related to address this has been incorporated into the manuscript on lines 241- 274.

2) Including a real-world application of the protocol - ideally with representative data, QC metrics, and phenotypic outputs - would greatly enhance accessibility for first-time users. This could be a brief case study embedded within the manuscript or a supplementary dataset linked to the methods.

An additional section has been added to the manuscript entitled '4. Case Study: Screening a *Klebsiella pneumoniae* Gene Deletion Library Against Vancomycin' for a real-world application of the protocol. It has been added in lines 552-611. For this case study we visualised our findings using ChemGenXplore, a Shiny application for

interactive visualisation of scored chemical-genomic datasets recently developed by our laboratory. It is available as a preprint and currently under review in the journal bioinformatics. As ChemGenXplore complements ChemGAPP (section 3.2) we decided to also add a small subsection (3.2.2) to describe its usage to the reader. We believe highlighting a web-based visualisation tool offers practical value to the reader, especially for those without prior knowledge in bioinformatics.

3) *While the protocol briefly mentions liquid culture and biofilm assays, this section could be expanded or linked more explicitly to existing protocols. Even a short paragraph highlighting necessary modifications (e.g., plate types, detection methods, time points) would broaden the scope and applicability.*

We acknowledge this point and have now addressed this from lines 661-669 of the manuscript as well as introducing a new section, 5.6 Adapting the Protocol for Liquid Culture Screening to cover this. We also cite the Brochado *et al.* 2018 paper for readers to find a good example of how to perform high-throughput liquid based culture based screens.

4) *A brief comparison in the introduction part between arrayed library-based screening and sequencing-based pooled screening strategies (e.g., Tn-seq, CRISPRi-seq) would help contextualize the advantages and limitations of this approach. This could include discussion of: Throughput vs. resolution, Cost, Scalability, Suitability for essential gene screening or strain-specific responses.*

This comparative introduction has been included in the main introduction of the manuscript from lines 48-64.

Section 1.1: The temperature unit should be formatted correctly as "5565 {degree sign}C" using an proper degree symbol. VWR one-well plates: It would be helpful to specify the dimensions (e.g., diameter) of the plates for reproducibility. Mycobacterium bovis BCG should be italicized.

Degree symbols have been amended and dimensions for the VWR one-well plates have been added. *Mycobacterium bovis* has been italicised.

Line 475: Replace "Bacto™" with "Bacto{trade mark, serif}" to correct the formatting of the trademark symbol.

Corrected.

Reviewer 2

General- Please clarify early in the manuscript that this protocol is tailored specifically for the pinning robot used in the authors' laboratory. Several steps may vary when using other robotic platforms, such as the Singer ROTOR. Acknowledging this explicitly will help readers adapt the protocol appropriately to their own equipment.

This clarification has been incorporated into manuscript from lines 118-126.

*Line 54 – References: the most recent reference from the biggest chemical genomics study in *S. cerevisiae* is missing, cite: Vieitez C, et al. Nature Biotech 2022 <https://pubmed.ncbi.nlm.nih.gov/34663920/> When mentioning Tong et al 2001 and Schuldiner et al 2005 mention that are high throughput screens but NOT chemical genomics.*

This reference has been added and cited in the manuscript on line 76.

Line 116- when talking about stresses, would be more accurate talking about conditions or clarity it further to avoid misunderstanding from the reader. For example heat stress is not “added” to the plate.

Re-edited the manuscript throughout so that stresses are now referred to as “conditions” to avoid misunderstanding.

Figure 2- wrong well in image (not A1, but H1)

This has been corrected.

Figure 2 at the end of the manuscript is not yet corrected

Line 121- Add a minimum number of hours need it until turning the plates upside down

This has been amended and the number of hours specified.

Figure 1- the image shows a source plate with 384 colonies, from the experience of this reviewer it would be better to replace the image for one with a couple of empty wells, since these empty wells are usually very informative as “contamination controls”.

This image has now been replaced to show the empty wells.

Figure 3B- “otherwise if the concentrations are too high Iris analysis will not work” – clarify in the text why Iris detection won’t work (minimum colony size? Too many empty wells?, any other issue?)

This section has been clarified further on lines 292-293.

Line 232- “plates are incubated based on the organism and stress condition” – link this text to Table 2 which provides further information.

This has been linked to Table 2.

Pretesting Challenges and Tips- Section 1.3 – “Consult literature data for conditions, examples such as ...” State the organism of each example to guide the reader to their favourite organism dataset.

This suggestion has been included alongside literature citations.

Line 307- Please clarify that the mention of non-dispensable pinning pads is specific to the particular robot model used in this protocol, as many laboratories use dispensable pinning pads in chemical genomics workflows.

This comment has now been addressed from lines 391-393.

Figure 6 – Consider increasing the clarity of this figure, if the starting point is a high density plate it is confusing why the output are few colonies.

A one line description of the figure addresses the reviewers' comments related to "why the output are few colonies".

*Line 349 - make sure the format is correct
Condition_Concentration_SourcePlate_Replicate (always “_” or interchange with “-”?)*

The format has been re-checked to ensure it is correct.

Line 379- “An alternative tool for processing growth curve data is the Kinetic Data Companion in R” this sentence is out of context since growth data was not mentioned in this protocol. I suggest to either put the sentence in context or just remove it.

The sentence has been removed for clarity.

Figure 8B- In the printed version of the manuscript, it is not visible that 8B shows stained colonies.

A clearer plate has been added that shows stained colonies.

We would like to thank you and the reviewers again for your constructive feedback. We hope the revised manuscript now meets the journal's standards and look forward to your consideration.

Yours sincerely,

Manuel Banzhaf and Georgia Willams
On behalf of all co-authors
Newcastle University

Response to Authors

I would like to thank the authors for addressing all comments, which help to clarify the mentioned aspects in the manuscript.

Only one Figure still needs to be corrected (Figure 2

Apologies for this – the image has now been corrected.

Response to Reviewers

Dear Julia Willett,

We would like to thank you and the reviewers for the constructive feedback and valuable suggestions provided on our manuscript. We have considered all comments and revised the manuscript accordingly. We believe that these revisions have increased the quality and clarity of our work.

We provide a detailed, point-by-point response to the reviewers' comments below.

For ease of review:

- **Reviewer comments** are shown in black *italicised* text.
- **Our responses** follow each comment in red text.
- Where changes/additions were made, we indicate the revised section(s) and line numbers in the manuscript.

Reviewer 1

1) *When transferring cells from library/source plates to screening plates, physiological variability between bacterial cultures could lead to artefactual results. It would strengthen the protocol to include recommendations on how to minimize this, such as:*

Synchronization of growth phase prior to spotting,

Consistent pre-culture conditions,

Use of internal plate controls or interleaved replicates,

Suggested metrics for assessing spotting consistency.

This point is acknowledged and a section related to address this has been incorporated into the manuscript on lines 241- 274.

2) *Including a real-world application of the protocol - ideally with representative data, QC metrics, and phenotypic outputs - would greatly enhance accessibility for first-time users. This could be a brief case study embedded within the manuscript or a supplementary dataset linked to the methods.*

An additional section has been added to the manuscript entitled '4. Case Study: Screening a *Klebsiella pneumoniae* Gene Deletion Library Against Vancomycin' for a real-world application of the protocol. It has been added in lines 552-611. For this case study we visualised our findings using ChemGenXplore, a Shiny application for

interactive visualisation of scored chemical-genomic datasets recently developed by our laboratory. It is available as a preprint and currently under review in the journal bioinformatics. As ChemGenXplore complements ChemGAPP (section 3.2) we decided to also add a small subsection (3.2.2) to describe its usage to the reader. We believe highlighting a web-based visualisation tool offers practical value to the reader, especially for those without prior knowledge in bioinformatics.

3) *While the protocol briefly mentions liquid culture and biofilm assays, this section could be expanded or linked more explicitly to existing protocols. Even a short paragraph highlighting necessary modifications (e.g., plate types, detection methods, time points) would broaden the scope and applicability.*

We acknowledge this point and have now addressed this from lines 661-669 of the manuscript as well as introducing a new section, 5.6 Adapting the Protocol for Liquid Culture Screening to cover this. We also cite the Brochado *et al.* 2018 paper for readers to find a good example of how to perform high-throughput liquid based culture based screens.

4) *A brief comparison in the introduction part between arrayed library-based screening and sequencing-based pooled screening strategies (e.g., Tn-seq, CRISPRi-seq) would help contextualize the advantages and limitations of this approach. This could include discussion of: Throughput vs. resolution, Cost, Scalability, Suitability for essential gene screening or strain-specific responses.*

This comparative introduction has been included in the main introduction of the manuscript from lines 48-64.

Section 1.1: The temperature unit should be formatted correctly as "5565 {degree sign}C" using an proper degree symbol. VWR one-well plates: It would be helpful to specify the dimensions (e.g., diameter) of the plates for reproducibility. Mycobacterium bovis BCG should be italicized.

Degree symbols have been amended and dimensions for the VWR one-well plates have been added. *Mycobacterium bovis* has been italicised.

Line 475: Replace "BactoTM" with "Bacto{trade mark, serif}" to correct the formatting of the trademark symbol.

Corrected.

Reviewer 2

General- Please clarify early in the manuscript that this protocol is tailored specifically for the pinning robot used in the authors' laboratory. Several steps may vary when using other robotic platforms, such as the Singer ROTOR. Acknowledging this explicitly will help readers adapt the protocol appropriately to their own equipment.

This clarification has been incorporated into manuscript from lines 118-126.

Line 54 – References: the most recent reference from the biggest chemical genomics study in S. cerevisiae is missing, cite: Vieitez C, et al. Nature Biotech 2022 <https://pubmed.ncbi.nlm.nih.gov/34663920/> When mentioning Tong et al 2001 and Schuldiner et al 2005 mention that are high throughput screens but NOT chemical genomics.

This reference has been added and cited in the manuscript on line 76.

Line 116- when talking about stresses, would be more accurate talking about conditions or clarity it further to avoid misunderstanding from the reader. For example heat stress is not “added” to the plate.

Re-edited the manuscript throughout so that stresses are now referred to as “conditions” to avoid misunderstanding.

Figure 2- wrong well in image (not A1, but H1)

This has been corrected.

Figure 2 at the end of the manuscript is not yet corrected

Apologies for this – the image has now been corrected.

Line 121- Add a minimum number of hours need it until turning the plates upside down

This has been amended and the number of hours specified.

Figure 1- the image shows a source plate with 384 colonies, from the experience of this reviewer it would be better to replace the image for one with a couple of empty wells, since these empty wells are usually very informative as “contamination controls”.

This image has now been replaced to show the empty wells.

Figure 3B- “otherwise if the concentrations are too high Iris analysis will not work” – clarify in the text why Iris detection won’t work (minimum colony size? Too many empty wells?, any other issue?)

This section has been clarified further on lines 292-293.

Line 232- “plates are incubated based on the organism and stress condition” – link this text to Table 2 which provides further information.

This has been linked to Table 2.

Pretesting Challenges and Tips- Section 1.3 – “Consult literature data for conditions, examples such as ...” State the organism of each example to guide the reader to their favourite organism dataset.

This suggestion has been included alongside literature citations.

Line 307- Please clarify that the mention of non-dispensable pinning pads is specific to the particular robot model used in this protocol, as many laboratories use dispensable pinning pads in chemical genomics workflows.

This comment has now been addressed from lines 391-393.

Figure 6 – Consider increasing the clarity of this figure, if the starting point is a high density plate it is confusing why the output are few colonies.

A one line description of the figure addresses the reviewers' comments related to "why the output are few colonies".

*Line 349 - make sure the format is correct
Condition_Concentration_SourcePlate_Replicate (always “ _ “ or interchange with “– “?)*

The format has been re-checked to ensure it is correct.

Line 379- “An alternative tool for processing growth curve data is the Kinetic Data Companion in R” this sentence is out of context since growth data was not mentioned in this protocol. I suggest to either put the sentence in context or just remove it.

The sentence has been removed for clarity.

Figure 8B- In the printed version of the manuscript, it is not visible that 8B shows stained colonies.

A clearer plate has been added that shows stained colonies.

We would like to thank you and the reviewers again for your constructive feedback. We hope the revised manuscript now meets the journal's standards and look forward to your consideration.

Yours sincerely,

Manuel Banzhaf and Georgia Williams
On behalf of all co-authors
Newcastle University

Re: mSystems00885-25R2 (**High-throughput chemical genomic screening: a step-by-step workflow from plate to phenotype**)

Dear Dr. Manuel Banzhaf:

Your manuscript has been accepted, and I am forwarding it to the ASM production staff for publication. Your paper will first be checked to make sure all elements meet the technical requirements. ASM staff will contact you if anything needs to be revised before copyediting and production can begin. Otherwise, you will be notified when your proofs are ready to be viewed.

Sincerely,
Julia Willett
Editor
mSystems